



# 1 Intra-regional transport of black carbon between the south edge of

# 2 North China Plain and Central China during winter haze episodes

Huang Zheng[1, 2], Shaofei Kong[1], Fangqi Wu[1], Yi Cheng[1], Zhenzhen Niu[1], Shurui Zheng[1], Guowei Yang[1], Liquan Yao[2],
Qin Yan[1,2], Jian Wu[1,2], Mingming Zheng[2,3], Nan Chen[3], Ke Xu[3], Yingying Yan[1], Dantong Liu[4], Delong Zhao[5], Tianliang
Zhao[6], Yongqing Bai[7], Shuanglin Li[1], and Shihua Qi[2]
[1]Department of Atmospheric Science, School of Environmental Sciences, China University of Geosciences, Wuhan, 430074,
China
[2]Department of Environmental Science and Technology, School of Environmental Sciences, China University of Geosciences,
Wuhan, 430074, China
[3]Hubei Provincial Environmental Monitoring Centre, Wuhan, 430072, China
[4]School of Earth, Atmospheric & Environmental Sciences, University of Manchester, M139PL, UK
[5]Beijing Weather Modification Office, Beijing, 100089, China
[6]School of Atmospheric Physics, Nanjing University of Information Science and Technology, Nanjing, 210044, China
[7]Hubei Key Laboratory for Heavy Rain Monitoring and Warning Research, Institute of Heavy Rain, China Meteorological
Administration, Wuhan, 430205, China
*Correspondence to*: Shaofei Kong (kongshaofei@cug.edu.cn)
**Abstract.** Black carbon (BC), from the incomplete combustion sources (mainly fossil fuel, biofuel and open biomass burning),
is chemically inertness and optical absorbance in the atmosphere. It has significant impacts on global climate, regional air
quality, and human health. During the transportation, its physical-chemical characteristics, optical properties and sources
would change dramatically. To investigate the BC properties (i.e., mass concentration, sources and optical properties) during
the intra-regional transport between the south edge of North China Plain (SE-NCP) and Central China (CC), simultaneous
observations of BC at a megacity (Wuhan, WH) in CC, three borderline cities (Xiangyang, XY, Suixian, SX and Hong'an, HA,
distributing from the west to east) between SE-NCP and CC and a city (Luohe, LH) in SE-NCP were conducted during the
typical winter haze episodes. Using Aethalometer, the highest equivalent BC (eBC) mass concentrations and aerosol absorption
coefficients ($\sigma_{abs}$) were found in the city (LH) at SE-NCP, followed by the borderline cities (XY, SX and HA) and megacity
(WH). The levels, sources, optical properties (i.e., $\sigma_{abs}$ and absorption Ångström exponent, AAE) and geographic origins of
eBC were different between clean and pollution episodes. Compared to clean days, the higher eBC levels (increased by
26.4−163%) and $\sigma_{abs}$ (increased by 18.2−236%) were found during pollution episodes due to more combustion of fossil fuel
(contributing for 51.1−277%), supported by the decreased AAE (by 7.40−12.7%). Non-parametric wind regression (NWR)



and concentration-weighted trajectory (CWT) results showed that the geographic origins of biomass burning (BC$_{bb}$) and fossil
fuel (BC$_{ff}$) combustion derived BC were different. Based on cluster analysis of trajectories, air parcels from south direction
dominated for border sites during clean days, with contributions of 46.0−58.2%, while trajectories from the northeast had
higher contributions (37.5−51.2%) during pollution episodes. At the SE-NCP site (LH), transboundary influences from south
direction (CC) exhibited more frequent impact (with the air parcels from this direction contributed 47.8% of all the parcels)
on the ambient eBC levels during pollution episodes. At WH, eBC was mainly from the northeast transport route during the
whole observation period. Two transportation cases showed that from upwind to downwind direction, the mass concentrations
of eBC, BC$_{bb}$ and BC$_{ff}$ all increased, while AAE decreased. This study highlighted that intra-regional prevention and control
for dominated sources of specific sites should be considered to improve the regional air quality.

## 1 Introduction

Black carbon (BC), a distinct type of carbonaceous material has attracted much attention mainly due to its climate effects over
past decades (Hansen et al., 2000; Jacobson, 2000; Bond et al., 2013). BC can strongly absorb but reflect less visible light and
the direct radiative forcing is estimated to be +0.88 W m$^{-2}$ (Bond et al., 2013). It is composed of small carbon spherules and
has large specific surface areas, which allows it to absorb aerosol, and provide substrate for atmospheric chemical reactions
(Liu et al., 2018a). BC also has adverse human health effects (Jansen et al., 2005; Cao et al., 2012) due to its absorption of
carcinogenic pollutants. Additionally, recent studies showed that BC can strongly impact the ambient air quality. For instance,
in urban areas, BC can enhance the haze pollution by modifying the planetary layer height, which was unfavorable to the
vertical dispersion of air pollutants (Ding et al., 2016). This "dome effect" is more substantial in rural areas under the same
BC conditions (Wang et al., 2018a). BC particle, coated with more materials can markedly amplify absorption and direct
radioactive forcing, which would further worsen the air quality (Peng et al., 2016; Liu et al., 2017a; Zhang et al., 2018). As
the transportation key points, the properties of BC at rural and urban sites needed to be emphasized, which is always ignored
in former field campaigns.
BC is formed only in combustion processes of carbon-based materials such as biomass and fossil fuels. The broadly reported
BC sources can be grouped into stationary sources (i.e., industrial emission), area sources such as residential coal/wood
combustion, open burning and mobile sources including diesel engines, etc. (Chow et al., 2011; Bond et al., 2013). To identify
the BC sources, several methods including aethalometer model, diagnosis ratios and radioactive carbon isotope have been
developed (Sandradewi et al., 2008; Verma et al., 2010; Zotter et al., 2016). Chow et al., (2011) summarized the ratios of
element carbon to PM$_{2.5}$ (expressed as percentage, %) from various sources and these ratios have been used to qualitatively
describe the BC sources (Liu et al., 2018a). Radiocarbon method can give the quantified results of BC sources as the
abundances of $^{14}$C/$^{12}$C in fossil fuel and modern carbon sources (i.e., biogenic sources) are different. Radiocarbon method
coupled with laevoglucose, a tracer of biomass burning have been adopted in BC source apportionment (Zhang et al., 2015a;
Liu et al., 2017b; Mouteva. et al., 2017; Salma et al., 2017). However, the technical limitations and the high cost for $^{14}$C



measurement block the application of radiocarbon method to BC source apportionment. The aethalometer model is an
alternative method, which can attribute the BC to fossil fuel combustion and biomass burning. The source apportionment can
be conducted using multi-wavelengths BC data (Sandradewi et al., 2008; Liu et al., 2018a) and the validity was proved by $^{14}C$
method (Zotter et al., 2016). Compared to other methods, aethalometer model can provide high-time resolution variation of
BC source contributions (Kalogridis et al., 2017; Liu et al., 2018a), which help to understand the atmospheric behaviors of BC,
especial for the temporal variations.
Atmospheric lifetime of BC varied from a few days to weeks and therefore, BC undergoes regional and intercontinental
transport (Bond et al., 2013). During the transport, its mixing state, morphology and optical properties will change (China et
al., 2015). As a result, BC has been observed in remote areas such as the polar regions (Huang et al., 2010; Weller et al., 2013;
Qi et al., 2017; Xu et al., 2017) and Tibetan Plateau (Cong et al., 2013). For instance, Qi et al., (2017) found that Asian
anthropogenic activities contributed 35−45% and biomass burning emissions from Siberian contributed 46−64% to the sources
of BC in Arctic in April 2008 by GEOS-Chem modeling. Similarly, Xu et al. (2017) also used the global transport model to
conclude that the anthropogenic emissions from eastern and southern Asia contributed most to the Arctic BC column loading
with percentages being 56% and 37% for spring and annual, respectively. To study the regional transport of BC, backward
trajectories and concentration-weighted trajectory (CWT) were always employed (Huang et al., 2010; Wang et al., 2017a). The
ratio of BC/CO ratio was adopted to study the BC aging during the transport processes (Verma et al., 2010; Pan et al., 2011;
Guo et al., 2017). Previous studies mostly focused on the impact of BC transportation on its physical-chemical properties
during aging at a given site (e.g., a megacity or a remote background site). A recent study indicated that in the south Ontario,
higher BC loading in the summer was partly from the trans-boundary fossil fuel derived BC emissions in the US (Healy et al.,
2017). To our knowledge, the interaction of BC transportation among various sites for a specific region has been rarely reported,
which may limit the understanding of regional-joint control for air pollution.
After continuous efforts, especially in the last five years, the spatial distribution pattern of air pollution has changed obviously
in China, and the positive result is that the average annual $PM_{2.5}$ concentration in Pearl River Delta (PRD) has achieved the
national secondary standard level (http://www.zhb.gov.cn/hjzl/zghjzkgb/lnzghjzkgb/). Now the key regions suffering from
severe $PM_{2.5}$ pollution are North China Plain (NCP), Yangtze River Delta (YRD), Sichuan Basin (SB), Fen-Wei River Basin
and Central China (CC). From Lin et al. (2018), it could be found that the air pollution areas at the south edge of North China
Plain (SE-NCP) and Central China were combined together and there existed obvious transportation routes between SE-NCP
and CC. The spatial distribution of aerosol optical depth (AOD) across China also verified that high values existed in Central
China (Tao et al., 2017). As an important chemical composition of $PM_{2.5}$, BC account for 7.1−25.3% (Huang et al., 2014) of
$PM_{2.5}$ mass. Despite large scale observation of ambient BC has been conducted (Table S1−S2), the BC studies are mainly
reported in NCP (Zhao et al., 2013; Ji et al., 2018; Liu et al., 2018b; Wang et al., 2017a), YRD (Zhuang et al., 2014, 2015,
2017), PRD (Cheng et al., 2008; Wu et al., 2009; Wang et al., 2017b) and Tibetan Plateau (TP) (Zhu et al., 2017; Niu et al.,
2018; Wang et al., 2018b). No studies have concerned the BC transportation and interaction between these key regions. BC
emission inventory suggests that different source categories exists between NCP and CC (Wang et al., 2014b; Qiu et al., 2016),




especially for the residential coal combustion (Qin and Xie, 2012). It should be emphasized that during the winter period, there
were central-heating activities in NCP, while no heating activities existed in Central China. It implied that the sources of BC
should be quite different. Therefore, the special geographic locations and terrain of Central China (Fig. 1) makes it complex
but provide an ideal opportunity to understand the BC levels, optical properties, sources and their variation during intra-
regional transportation between the two polluted regions. However, corresponding researches have not been reported.
Therefore, the aims of this study were to (1) study the differences of BC levels, sources, and optical properties under different
air pollution situation at this region; (2) quantify the regional transportation of BC at multiple observation sites in CC and SE-
NCP. To study the BC sources, the diagnosis ratios and aethalometer model were used. The backward trajectory based methods
were employed to quantify the potential regional transport contribution. This paper firstly reported the sources of BC in Central
China and gave the direct evidence of BC properties variation during the regional transportation between two key regions of
China, which is helpful to develop effective countermeasures for mitigating regional air pollution.

## 2 Methodology

### 2.1 Observation plan

For the sites selection, we refereed to the trajectory of air masses reached to Wuhan for January 2017 (Figure S1) and found
that the north and northwest direction were the main directions. For the north direction, the air masses originated at the SE-
NCP and Luohe is just on the north routes and close to the origination region as the Figure 1 shown. Meanwhile, the Central
China is not an isolated region. From Figure 1, there were two obvious connection channels for PM$_{2.5}$ between SE-NCP and
CC, which was decided by the mountain crossing the two regions. Therefore, to investigate the regional transport of air
pollutants and also answer whether the pollutants in CC can be transported to SE-NCP in winter, five sites including WH, three
borderline cities (Xiangyang, XY; Suixian, SX and Hong'an, HA, distributing from the west to east) between NCP and CC
and a city (Luohe, LH) in SE-NCP were selected as shown in Figure 1. The observation site at Wuhan locates in the Hubei
Environmental Monitoring Centre, which is an urban site with no industrial emission sources. LH and XY sites are located in
the suburban areas. HA and SX sites belong to rural areas. The observation instruments were placed near the local environment
monitor stations and therefore, the six routine-monitored air pollutants including PM$_{2.5}$, PM$_{10}$, NO$_2$, SO$_2$, CO and O$_3$ were
available. Black carbon measurement instruments including Magee Scientific-AE31, AE33 and AE51 were deployed (Table
1). The observation periods started from 8[th] January after a regional snowfall event and ended at 25[th] January 2018 before
another snowfall event coming. The observation duration at the five sites are summarized in Table 1.

### 2.2 Instrument description

AE31 continuously collects the ambient BC on the quartz tape and measures the light singles on sampled spot ($I$) and
reference spot ($I_0$) and the light attenuation (ATN) is defined as:



$ATN = -100ln(\frac{I}{I_0})$       (1)
It assumes the relation between BC mass loading and the delta of ATN as a result of BC deposited on the tape is linear. The
BC mass concentration is calculated as following:
$BC = \frac{d(ATN)}{MAC} \times \frac{A}{V}$       (2)
where MAC is the mass specific attenuation cross section (m$^2$ g$^{-1}$); $A$ is the area of sampled spot (1.67 cm$^2$); $V$ is the volume
of the sampled air passing through the tape. The disadvantage of AE31 is the filter loading effect, which needs further
correction to compensate (Petit et al., 2015). The BC absorption coefficient ($b_{abs}$, Mm$^{-1}$) is calculated as:
$b_{abs} = \frac{BC \times MAC}{C \times R(ATN)}$       (3)
where $C$ is the calibration factor (2.14 for quartz material tape); $R$ ($ATN$) is a correction factor for shadowing effect and it is
empirically determined using the compensation parameter $f$ (Weingartner et al., 2003):
$R(ATN) = \left(\frac{1}{f} - 1\right)\frac{\ln(ATN)-\ln(10)}{\ln(50)-\ln(10)} + 1$       (4)
To overcome the shortage of loading effect, AE33 (dual spot) was developed. It also simultaneously measures the ATN at
seven wavelengths. Different to AE31, AE33 measures BC on two parallel spots on the fibre tape (Teflon-coated) with different
flow rate:
$BC_1 = BC \times (1 - k \cdot ATN_1)$       (5)
$BC_2 = BC \times (1 - k \cdot ATN_2)$       (6)
The loading compensation $k$ is calculated according the equation (5) and (6) and the BC mass concentration is further calculated
as following:
$BC = \frac{A[d(ATN)/100]}{F_1(1-\varphi)MAC \cdot C(1-k\ ATN_1)dt}$       (7)
For AE33, the area of sampled spot (A) is 0.785 cm$^2$ and enhancement parameter (C) is 1.57 for Teflon-coated fibre. The
absorption coefficient ($\sigma_{abs}$, Mm$^{-1}$) by AE33 is estimated as multiplying BC mass concentration by MAC.More details about
the BC concentration calculation, parameters (i.e., $f$ and MAC for different wavelengths) and the differences between AE31
and AE33 can be found in previous study (Rajesh and Ramachandran, 2018). AE51 measures the absorbance (ATN) of the
loaded spot (3 mm diameter) and the reference portion of the Teflon-coated borosilicate glass fiber using a stabilized 880 nm
LED light source. The flow rate of AE51 was set as 100 mL min$^{-1}$ and more information about AE51 can be found online
(https://aethlabs.com/microaeth/ae51/tech-specs).
**2.3 Data processing**
**2.3.1 BC source apportionment**
BC absorbs the solar spectrum efficiently with a weak dependence on wavelength and the absorption Ångström exponent
(AAE) is used to describe this spectral dependence of absorption (Zhu et al., 2017). The AAE value varies significantly from



one source to another, i.e., the AAE values for fossil fuel combustion and biomass burning derived BC are 1.0 and 2.0,
respectively (Sandradewi et al., 2008). Therefore, a BC source apportionment method was established based on the AAE
(Sandradewi et al., 2008) and was verified by [14]C method (Zotter et al., 2016).
Black carbon source apportionment using aethalometer model is based on the assumption that the aerosol absorption coefficient
is from fossil fuel combustion derived BC ($BC_{ff}$) and biomass burning derived BC ($BC_{bb}$). Because the absorption coefficients
at different wavelengths are different and the absorption of $BC_{ff}$ and $BC_{bb}$ follow different spectral dependencies. The Ångström
exponents: $\alpha_{ff}$ and $\alpha_{bb}$ are used to describe the dependencies of fossil fuel and biomass burning, respectively. The following
equations are used in BC source apportionment (Sandradewi et al., 2008):
$$\frac{b_{abs}(470nm)_{ff}}{b_{abs}(950nm)_{ff}} = \left(\frac{470}{950}\right)^{-\alpha_{ff}} \tag{8}$$
$$\frac{b_{abs}(470nm)_{bb}}{b_{abs}(950nm)_{bb}} = \left(\frac{470}{950}\right)^{-\alpha_{bb}} \tag{9}$$
$$b_{abs}(470nm) = b_{abs}(470nm)_{ff} + b_{abs}(470nm)_{bb} \tag{10}$$
$$b_{abs}(950nm) = b_{abs}(950nm)_{ff} + b_{abs}(950nm)_{bb} \tag{11}$$
$$BB(\%) = \frac{b_{abs}(950nm)_{bb}}{b_{abs}(950nm)} \tag{12}$$
where $b_{abs}$(470 nm) and $b_{abs}$(950 nm) are the black carbon absorption coefficients at 470 and 950 nm wavelengths, respectively.
Due to the single channel (λ=880 nm) of AE51, BC source apportionment results were not available at SX and XY and were
only reported for HA, LH and WH.

## 2.3.2 Assessment of surface transport

Generally, the north wind dominated in winter in CC and air pollutants can be transported from upwind direction (north) to
downwind direction (south). In order to evaluate the effects of regional transport, the surface transport under specific wind
direction and speed per unit time was calculated according to the previous study (Wang et al., 2018b):
$$f = \frac{1}{n}\sum_{i=1}^{n} C_i \times WS_i \times cos\theta_i \tag{13}$$
where $f$ stands for the surface flux intensity of BC (μg s$^{-1}$ m$^{-2}$); $n$ is the sum of observation hours; $WS_i$ and $C_i$ stand for the
hourly average of wind speeds (m s$^{-1}$) and BC mass concentrations (μg m$^{-3}$) in the $i$th observation duration, respectively; $\theta_i$
represents the angle differences between hourly wind direction and the defined transport directions (i.e., northwest-southeast
for HA, SX and WH and north-south direction for LH and XY).

## 2.4 Potential geographic origins

The concentration-weighted trajectory (CWT) is always used to assess the regional transport of air pollutants (Kong et al.,
2018; Zheng et al., 2018). This method is based on backward trajectory analysis. Prior to CWT analyses, the backward
trajectory calculating was firstly conducted in each sampling site. The input wind datasets into HYSPLIT are downloaded from
the Nation Oceanic Atmospheric Administration (NOAA) (ftp://arlftp.arlhq.noaa.gov/pub/archives/gdas1/). For backward



trajectory analysis, the air masses reaching at each observation site during the sampling period were calculated for 24 times
with 1-hour resolution each day (starting from 0:00 to 23:00) at 200 m AGL (Fig. S2). These trajectories were than clustered
according to their geographic origins (Fig. 1). For CWT analysis, a user-friendly tool Zefir written in Igor was used (Petit et
al., 2017a). The domain covered by trajectories was divided into thousands of cells with 0.2º × 0.2º. More description about
CWT can be found in text S1.
**2.5 Auxiliary dataset**
Hourly meteorological dataset including sea level pressure, temperature, relative humidity, wind speed, wind direction and
visibility were acquired form the China Meteorological Data Service Centre (CMDC) (http://data.cma.cn, last accessed:
2018/1/26). The every 3-hour boundary layer height (BLH) was acquired from the NOAA's READY Archived Meteorology
online calculating program (http://ready.arl.noaa.gov/READYamet.php, last accessed: 2018/4/8). Figure S3 shows the hourly
averaged meteorological parameters at the five sites. The meteorological conditions at the five sites followed the same variation
trends. However, significant differences ($p < 0.01$) of these parameters were found (Table S3). For instance, the average
pressure, temperature and relative humidity at WH were significant higher $(p < 0.01)$ than those at LH. For BLH, the mean
values of the five sites showed insignificant differences. .
As mentioned in section 2.1, the five observation sites were located next to the local air quality monitoring stations and
therefore, six air pollutants ($PM_{10}$, $PM_{2.5}$, $SO_2$, $NO_2$, CO and $O_3$) were available and the data were downloaded from the China
Environmental Monitoring Centre (http://www.cnemc.cn, last accessed: 2018/4/10). Figure S4 shows the hourly average of
six air pollutants at the five sites during the observation periods. The major air pollutant was $PM_{2.5}$ during the entire research
campaign. According to the Ambient Air Quality Standards (GB3095-2012), the air quality can be classified into clean, light
polluted and heavily polluted when $PM_{2.5}$ mass concentrations are less than 75, between 75−250 and greater than 250 μg m$^{-3}$,
respectively. Similar air quality classification was also reported elsewhere (Zheng et al., 2015; Zhang et al., 2018). Detailed
information about the daily air quality of each site is shown in Fig. S5.
**3 Results and discussion**
**3.1 General characteristics**
Time series and box plots of the eBC concentrations (measured at 880 nm) at the five sites are shown in Fig. 2. The highest
eBC concentrations were observed at LH (8.48 ± 4.83 μg m$^{-3}$), followed by XY (7.35 ± 3.45 μg m$^{-3}$), HA (5.54 ± 2.59 μg m$^{-3}$),
SX (4.47 ± 2.90 μg m$^{-3}$), and Wuhan (3.91 ± 1.86 μg m$^{-3}$). Despite the sampling periods, site types, inlet of aerosol and
instruments were different between different studies (Table S1), BC was generally higher in North China and lower BC levels
were found in remote areas and coastal areas. From BC emission inventory, North and Central China hold higher BC emission
intensity (Qin and Xie, 2012; Yang et al., 2017), i.e., emission rates in Hubei and Henan provinces were about 0.6−1.0 g C m$^{-2}$



yr$^{-1}$, which were higher than other regions (Yang et al., 2017). Simulation results also suggested that near-surface
concentrations of BC (6−8 µg m$^{-3}$) in Hubei and Henan were higher than those in south China (4−6 µg m$^{-3}$) during winter
(Yang et al., 2017). Compared to the results in other countries, BC levels in this study were higher than those in Finland
(Hyvärinen et al., 2011), France (Petit et al., 2017b), Ontario (Healy et al., 2017), and south Africa (Chiloane et al., 2017).
For the aerosol absorption properties measured at seven wavelengths by aethalometer, the characteristics (i.e., temporal
variation) are generally consistent with each other and the corresponding properties for wavelength at 520 nm is mostly
discussed (Zhuang et al., 2015, 2017; Wang et al., 2017b). Therefore, we only discussed the absorption properties at $\lambda$ = 520
nm. Figure 3 shows the frequency distribution of absorption coefficients ($\sigma_{abs}$) at various sites. $\sigma_{abs}$ measured at HA, LH, and
WH exhibited a single peak pattern. The average values of $\sigma_{abs}$ measured at HA, LH and WH were 86.0, 132 and 60.6 Mm$^{-1}$,
respectively. Similar to the spatial distribution of BC levels, higher $\sigma_{abs}$ was found in North and Central China, while lower
values observed in coastal areas and Tibetan Plateau (Table S2).
Figure 3 also shows the average absorption spectra measured at seven wavelengths for different sites. The power law fit was
used to calculate the AAE (Zhu et al., 2017). The highest average AAE value was found at LH (1.40), followed by WH (1.34)
and HA (1.32). The results indicated that the AAE was different at urban, suburban and rural sites. Generally, the AAE from
coal combustion (2.11−3.18) (Sun et al., 2017) and biomass burning (1.85−2.0) (Petit et al., 2017b) were higher than that from
traffic sources (0.8−1.1) (Sandradewi et al., 2008; Olson et al., 2015). Therefore, AAE in different sites suggested the different
energy consumption structure and more coal or biomass were burned in North China (i.e., LH in this study).

**3.2 Clean days *vs* pollution episodes**

Figure. 4 shows the eBC concentrations under different air pollution situation. It clearly shows that the eBC concentrations
increased as the deterioration of air quality. For instance, at LH, the average eBC concentrations were 3.39 ± 2.06 µg m$^{-3}$, 8.31
± 4.55 µg m$^{-3}$ and 13.0 ± 4.59 µg m$^{-3}$, respectively when the air quality was clean, light polluted and heavy polluted. The
average values of eBC increased by 163%, 139%, 96.2%, 51.8% and 26.4% at SX, XY, LH, HA and WH, respectively from
clean to pollution. The enhancement of eBC along with the deterioration of air quality was also reported elsewhere (Wang et
al., 2014a; Liu et al., 2016, 2018b). At LH and HA, the enhancement eBC level from clean to pollution period was due to both
the elevated BC emissions from biomass burning (BC$_{bb}$) and fossil fuel combustion (BC$_{ff}$) (Fig. 4b and 4c). The BC$_{ff}$ accounted
for a higher contribution to eBC and the percentages of BC$_{bb}$ to eBC decreased during the haze episodes (Fig. 4d). At WH, the
concentration and percentage of BC$_{bb}$ both decreased from clean to pollution, which suggested that more BC$_{ff}$ was emitted
during haze episodes. This finding was different with previous study conducted in Beijing that the percentage of BC$_{bb}$ to eBC
increased from clean to pollution episodes (Liu et al., 2018b). The differences suggested that the control of fossil fuel
combustion (vehicle emissions) instead of coal or biomass burning should be taken priority during the haze episodes in WH.
While it should give priority to biomass and coal combustion control in North China to prevent air pollution.
Additionally, the aerosol optical properties ($\sigma_{abs}$ and AAE) also exhibited different levels under different air pollution situation.
Similar to eBC levels, the $\sigma_{abs}$ also elevated by 11.7−254% as the air quality switched from clean to pollution (Fig. 4e). There





are more secondary aerosols (i.e., sulfate, nitrate) during the pollution episodes (Huang et al., 2014), and the increased
secondary aerosols would be more adsorbed on the surface of BC. The absorption of BC therefore enhanced via the lens effects
of these coated materials (Jacobson, 2000; Moffet and Prather, 2009). On the contrary, the AAE showed higher values during
clean days compared to pollution episodes (Fig. 4f). The decreasing of AAE from clean to polluted days was also reported
elsewhere (Zhang et al., 2015b) and it can be partly attributed to the source variation. For instance, the AAE for biomass
burning is about 2.0 while the AAE for fossil fuel combustion is about 1.0 (Sandradewi et al., 2008). Higher AAE values
during clean days suggested that more BC was from biomass burning and lower AAE indicated the dominance of fossil fuel
combustion during the pollution period (Fig. 4c).
Figure 5 shows the diurnal variations of eBC under different air quality. At HA, LH and SX, after sunrise, an increasing and a
peak value at about 09:00 local time (LT) were observed. This variation was more obvious during pollution days due to the
higher eBC levels. The morning peak may be related with the combined effects of increased biomass burning and fossil fuel
combustion emissions (Fig. S6). Additionally, the low mixing layer height in the morning also favored the accumulation of
eBC. After sunrise, with the elevation of BLH, the eBC levels decreased and the minimum occurred at about 15:00 (LT). In
the evening hours, eBC showed increasing trends and peaks at about 21:00. The same diurnal patterns of eBC were also
reported in other areas (Verma et al., 2010; Ji et al., 2017; Liu et al., 2018b). However, the diurnal variations of eBC at WH
and XY exhibited different patterns during clean or pollution episodes. The diurnal pattern of eBC at WH was not likely
controlled by the development of mixing layer height, which may lead to the maximum and minimum values of air pollutants
generally occurring at sunrise and afternoon, respectively. The unexpected lower value in the morning (about 09:00 LT) and
higher value in the afternoon (15:00), at WH needs further research.
**3.3 Ratios of BC/PM$_{2.5}$ and BC/CO**
The BC/PM$_{2.5}$ and BC/CO ratios are widely used to identify the BC sources (Zhang et al., 2009; Wang et al., 2011; Verma et
al., 2010; Chow et al., 2011). Generally, the ratios of BC/PM$_{2.5}$ from oil combustion (traffic) and agricultural burning are higher
than those from industrial emissions such as manufacturing and mineral products. For instance, the mobile sources hold the
highest ratios of BC/PM$_{2.5}$ (0.33−0.77) and the residential wood combustion shows lower ratio (0.056) (Chow et al., 2011).
For the BC/CO ratios (μg m$^{-3}$/ppbv), it also varied from different sources, i.e., traffic (0.0052), industry (0.0072), power plant
(0.0177), and residential (0.0371) (Zhang et al., 2009). In this study, the BC, PM$_{2.5}$ and CO were well correlated with each
other (Fig. S7). The correlation coefficients ($r^2$) between BC and PM$_{2.5}$ were 0.67, 0.30, 0.44, 0.37 and 0.48 at LH, HA, WH,
SX and XY, respectively. Significant correlations ($p < 0.05$) between BC and CO were found with $r^2$ ranging from 0.27 (XY)
to 0.71 (LH). The good correlations indicated that the BC, PM$_{2.5}$ and CO may be from similar sources (except HA with low $r$
value as 0.06).
Overall, BC in this study was not likely from industrial emissions (Fig. 6a), as the BC/PM$_{2.5}$ ratios (μg m$^{-3}$/μg m$^{-3}$) in this
study (0.045−0.083) were  higher than those from industry (0.0046−0.03) (Chow et al., 2011). Instead, BC/PM$_{2.5}$ ratios at the
five sites were all within the range of oil combustion (0.03−0.136). Additionally, the BC/PM$_{2.5}$ ratios at LH and SX were in



line with the ratio from residential wood combustion. From BC/CO ratios, BC was more likely from biomass burning (crop
residue: 0.0056−0.016) at HA and LH, while it was mainly from gasoline combustion in SX, WH, and XY (Fig. 6b). Quantified
calculation results using equations in section 2.3.1 also suggested that the fractions of BC from biomass burning at HA (27.6
± 9.40%) and LH (29.5 ± 9.14%) were significant higher ($p < 0.01$) than that at WH (25.4 ± 11.8%). Compared to other urban
areas, the ratios of BC/CO (µg m$^{-3}$/ppbv) at SX (0.004), and WH (0.0044) were lower than those in Beijing (0.0058) (Han et
al., 2009), Guangzhou (0.0054) (Verma et al., 2010), Gwanjun (0.006) (Park, 2005) and Tokyo (0.0057) (Kondo et al., 2006)
as well as Mt. Huang (0.0065) (Pan et al., 2011), while ratios at HA (0.0091) and LH (0.0076) were higher than these studies.
Due to the longer atmospheric lifetime of CO, the ratio of BC to CO and the correlation coefficients of them would decreased
from upwind to downwind directions (Guo et al., 2017). Therefore, the BC/CO is used to reflect the BC aging during the
transport (McMeeking et al., 2010; Verma et al., 2010; Guo et al., 2017). Table 2 summarizes the BC/CO ratios at different
observation sites under different air pollution situation. Except for WH, clean days held significant lower BC/CO ratios than
those during the pollution episodes ($p < 0.01$). For instance, at LH, the BC/CO ratio during clean days (0.0058 ± 0.0024 µg
m$^{-3}$/ppbv) was significant lower ($p < 0.01$) than heavy pollution episodes (0.0071 ± 0.0013 µg m$^{-3}$/ppbv). The coefficients ($r^2$)
also decreased from pollution (0.52) to clean days (0.27). The BC/CO ratios suggested that when the air quality was good, BC
was more aged while it was fresher during the pollution episodes.

### 3.4 BC under different wind direction and speed

Nonparametric wind regression (NWR) was used to identify and quantify the impact of likely source regions of air pollutants
as defined by wind direction and speed (Henry et al., 2009). Fig. S8 shows the eBC levels under different wind speed and
directions at the five sites. As shown in Fig. 1, SX and HA are located in the northwest direction of WH and therefore, high
eBC levels were found in the northwest directions of SX, HA and WH when north wind dominated. On the contrary, when the
south wind dominated, BC was blowing from south to the north direction and high levels were found in the south direction at
WH and HA. However, at LH and XY, higher levels of BC were only found from south direction. In addition to eBC levels,
the BC$_{bb}$ and BC$_{ff}$ under different wind speed and directions were also discussed at HA, LH and WH (Fig. 7). At HA, the NWR
plots of BC$_{ff}$ was in line with eBC and high levels were from both northwest and south directions while the high level of BC$_{bb}$
was only found in southeast direction. The same result was also found at WH. High level of BC$_{bb}$ was due to more biomass
burning in the southeast direction of HA and WH. At LH, the NWR plots of BC$_{bb}$ and BC$_{ff}$ were the same with the eBC as
discussed above.
In order to describe the BC transportation from upwind to downwind directions, we used Eq. (6) in section 2.3.3 to calculate
the surface transport of eBC (Fig. 8). The calculated average SAT values of BC were −0.69 ± 10.2, −0.06 ± 12.0, −0.17 ± 5.33,
0.29 ± 6.14 and 0.99 ± 17.8 µg s$^{-1}$ m$^{-2}$, respectively for HA, LH, SX, WH and XY. The negative values at HA, LH and SX
suggested that the transportation intensity of BC from south (southeast) to north (northwest) direction was higher, while the
positive values observed at WH and XY indicated that more BC was transported from north direction to south direction. The





large standard deviation of SAT reflected strong fluctuations in transport, which was due to wind speed, directions and BC
levels (Wang et al., 2018b).

**3.5 Potential geographic origins**

Employing CWT method, the potential geographic origins of eBC for the five sites were explored (Fig. S9). Overall, CWT
results of eBC at the five sites suggested that high eBC levels were found both in the north and south directions of LH and
WH, while the high levels (i.e., > 4 μg m$^{-3}$) of eBC were only found from northeast directions of HA, SX and XY (Fig. S9).
In addition to the possible geographic origins of eBC, the source regions of $BC_{bb}$ and $BC_{ff}$ at HA, LH and WH were also
discussed. At HA, the CWT results showed that high levels of eBC (i.e., > 3 μg m$^{-3}$) were from north /northeast direction.
However, the hot spots of $BC_{bb}$ and $BC_{ff}$ were different, with higher levels of $BC_{bb}$ from both south and north directions and
higher levels of $BC_{ff}$ from the north direction. The hot spots of $BC_{bb}$ and $BC_{ff}$ occurring in the north of HA were likely due to
the intensive BC emission in this area. Also, higher levels of $BC_{bb}$ and $BC_{ff}$ were found in the south of LH. Opposite to the
CWT results at HA, the hot spots of $BC_{bb}$ was only found in the southeast direction of WH and high levels of $BC_{ff}$ were found
in the north and south directions of WH. The CWT results at WH were in line with the wind rose plots in section 3.4. The unity
of CWT and wind rose plots at WH suggested that there were intensive biomass burning activities in the south direction of
WH during the observation period, which was verified by the MODIS fire-points distribution (Fig. S10).
Furthermore, we also discussed the source region differences of eBC under different air quality (Fig. 10). The higher levels
(>1 μg m$^{-3}$) of eBC were mainly from the south direction of HA when the air was clean. While during the pollution episodes,
air parcels from the north direction contributed high concentrations (> 3 μg m$^{-3}$) to the BC at HA. The CWT plots of $BC_{bb}$ and
$BC_{ff}$ showed similar distribution with eBC. At LH, CWT results of eBC, $BC_{bb}$ and $BC_{ff}$ showed that high levels of eBC were
mainly from south direction during clean days. When the air quality degraded to pollution, the air masses from south and east
directions both contributed to the high eBC (i.e., > 5 μg m$^{-3}$ for BC) at LH. CWT results at WH showed that the southeast
direction was the dominant source regions of eBC, $BC_{bb}$ and $BC_{ff}$ during clean days, while the source regions switched to
northeast direction when the air quality changed into pollution.
To give quantified results of which cluster had greater contribution to eBC level at the receptor sites, the percentage
contributions of each cluster reaching at the five sites under different air pollution situation are summarized in Figure 11.
Trajectories from south was the main transport pathways reaching at HA, which accounted for 49.6% for clean days and
therefore, the highest average eBC level (6.33±1.61 μg m$^{-3}$) was found from south direction during the clean days. However,
the percentage contribution for the south cluster contributed least (21.4%) to the total air masses, but it had the highest eBC
level (7.89±2.59 μg m$^{-3}$) among the three clusters during the pollution episode. At LH, despite the lowest ratios of trajectories
were found from south (16.7%) and northwest (21.6%), respectively for clean and pollution days, these two clusters had the
highest eBC levels. At WH, the cluster with the highest percentages also had the highest BC levels. For instance, northeast
direction was the primary pathway of BC reaching at WH and the highest average eBC value was also found from this direction.



In summary, at the boundary sites (HA, SX and XY), BC was mainly from south direction (accounting for 46.0−58.2%) when
the air quality was clean and it was mainly from northeast/northwest directions (51.2−76.5%) when the air quality getting
worse. At SE-NCP site (LH), BC was dominantly from south direction (47.8%) during pollution episodes in this study. At CC
site (WH), BC was mainly from northeast direction (49.3−71.1%). These results suggested that northwest and northeast
directions were the main transport pathways of air pollutant reaching to WH during the pollution episodes. Furthermore, in
addition to control local emissions during haze episodes, the emission sources, i.e., industry plant and open biomass burning
in the upwind direction should also be controlled to prevent the further deterioration of air quality.
**3.6 Case studies for BC properties variation during transportation**
To explore the BC variations (i.e., mass concentration, sources and AAE) during the transportation, we chose two cases. LH
and HA were selected as the study sites due to the same instrument deployment (AE33) and they are representative of SE-NCP
and CC. BC transportation from HA to LH and from LH to HA were both considered. Figure 12a shows the hourly backward
trajectories reaching at HA on 2018-1-12 and the trajectory at 13:00 (UTC) was found passing through LH (black line) and the
travelling time was about 28 h. Therefore, the eBC mass concentration (including $BC_{ff}$ and $BC_{bb}$) and AAE at the upwind site
LH on 8:00 2018-1-11 and downwind site HA on 13:00, 2018-1-13 (UTC) were compared. During the air transport from LH
to HA, eBC, $BC_{ff}$ and $BC_{bb}$ significantly increased ($p < 0.01$), while the AAE significantly decreased from $1.49 \pm 0.02$ to $1.42$
$\pm 0.02$ ($p < 0.01$). Similarly, in case 2, the air masses reaching at LH on 7:00, 2018-1-13 (UTC) were also passing through HA
before about 31 h ago (black line) (Figure 12c). The eBC and $BC_{ff}$ increased from upwind (HA) to downwind (LH), while
$BC_{bb}$ and AAE decreased from $2.37 \pm 0.23$ μg m$^{-3}$ and $1.43 \pm 0.02$ to $12.14 \pm 0.14$ μg m$^{-3}$ and $1.32 \pm 0.01$, respectively.
The eBC mass concentrations were found enhanced during the transportation regardless of the transport direction was from
CC to NCP or from NCP to CC. Previous studies found that the BC coagulation with non-refractory materials becomes more
significant when aging timescale greater than 10 h (Riemer et al., 2004). In these two cases, the travelling time (aging time)
from LH to HA and from HA to LH were 28 h and 31 h, respectively, which suggested that the BC particle should be coagulated
through complex atmospheric processes. Therefore, the new emission inputs along the trajectory enhanced the eBC mass
concentration during the transport. However, slight differences found for $BC_{bb}$ transport: $BC_{bb}$ increased from LH ($1.28 \pm 0.06$
μg m$^{-3}$) to HA ($2.57 \pm 0.47$ μg m$^{-3}$), while $BC_{bb}$ decreased from HA ($2.37 \pm 0.23$ μg m$^{-3}$) to LH ($2.14 \pm 0.14$ μg m$^{-3}$). On the
contrary, the AAE values were found decreased during the transport. The same result was also found through numerical
simulation (Liu et al., 2018b). Additionally, chamber study of diesel soot particles coated with secondary organic compound
also found that the AAE decreased from 1.13 to 0.8 (Schnaiter, 2005). The decreasing of AAE of BC particles during the
transport indicated that BC was coated by other materials.



## 4 Summary

In order to understand the levels, optical properties, sources, regional transportation and aging of BC in Central China and south edge of North China Plain during winter haze episodes, simultaneous observations at rural sites (HA and SX), suburban (LH and XY) and megacity (WH) were conducted during January 2018. Using the diagnosis ratios, aethalometer model, backward trajectory and concentration-weighted trajectory (CWT) methods, conclusions were drawn as following:

(1) Generally, the highest ambient eBC was found in northern sites ($8.48 \pm 4.83$ µg m$^{-3}$ and $7.35 \pm 3.45$ µg m$^{-3}$ at LH and XY), followed by the transport route sites ($5.54 \pm 2.59$ µg m$^{-3}$ and $4.47 \pm 2.90$ µg m$^{-3}$ for and HA and SX), and southern site ($3.91 \pm 1.86$ µg m$^{-3}$ for WH).

(2) Levels, sources, optical properties, and diurnal variation of eBC were different under different air quality. eBC concentrations and absorption coefficients ($\sigma_{abs}$) increased by 26.4−163% and 11.7−254% respectively, from clean to pollution episodes. The increasing may due to more fossil fuel combustion emissions during pollution episodes, supported by lower Ångström exponent (AAE) and higher BC$_{ff}$ concentrations.

(3) BC/PM$_{2.5}$ and BC/CO ratios suggested that BC was mainly from oil combustion and residential wood or biomass combustion in this region. The higher BC/CO ratios during pollution episodes than those for clean days suggested that the BC particle was fresher during pollution days. (4) Nonparametric wind regression (NWR) results of BC$_{bb}$ and BC$_{ff}$ showed different dominate source regions with BC$_{bb}$ mainly from southeast direction and BC$_{ff}$ from both northwest and southeast of WH and HA. However, BC$_{bb}$ and BC$_{ff}$ were mainly from south direction of LH.

(5) At the boundary sites (HA, SX and XY), eBC was dominantly from south direction (accounting for 46.0−58.2%) when the air quality was clean and it was mainly from northeast/northwest directions (51.2−76.5%) during pollution episodes. At the SE-NCP site, air masses from south direction accounted for 47.8% of ambient BC level when the air was polluted. At the CC site, air parcels from northeast contributed 49.3−71.1% to the BC loading during the entire observation period (i.e., clean days and pollution episodes).

(6) During the transportation from upwind to downwind direction, BC mass concentration increased, while the AAE decreased. This study firstly revealed the differences of levels, optical properties and sources of BC at five sites in south edge of North China Plain and Central China during winter haze episodes and discussed the interaction of BC between two key polluted regions. It was expected to be a demonstration for corresponding researches on regional interaction of BC transportation during winter haze episodes for other regions.

*Data availability.* Data is available on request to kongshaofei@cug.edu.cn.

*Acknowledgement.* This study was financially supported by the Key Program of Ministry of Science and Technology of the People's Republic of China (2016YFA0602002; 2017YFC0212602), the Key Program for Technical Innovation of Hubei Province (2017ACA089) and the Program for Environmental Protection in Hubei Province (2017HB11). The research was




also funded by the Start-up Foundation for Advanced Talents (201616) and the Fundamental Research Funds for the Central
Universities (201802), China University of Geosciences, Wuhan.

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



**Table 1** Information of the observation sites, periods and instruments

| Sampling site | Location | Site type | Sampling period | Instrument | Data resolution |
|---|---|---|---|---|---|
| Hong'an (HA) | 114.58º E, 31.24º N | Rural | 1/8 13:00~1/25 9:00, 2018 | AE33 | 1-minute |
| Luohe (LH) | 114.05º E, 33.57º N | Suburban | 1/9 18:00~1/25 9:00, 2018 | AE33 | 1-minute |
| Suixian (SX) | 113.28º E, 31.88º N | Rural | 1/10 09:00~1/25 8:00, 2018 | AE51 | 1-minute |
| Wuhan (WH) | 114.39º E, 30.53º N | Urban | 1/8 15:00~1/25 8:00, 2018 | AE31 | 5-minute |
| Xiangyang (XY) | 112.17º E, 32.02º N | Suburban | 1/10 09:00~1/25 8:00, 2018 | AE51 | 1-minute |



**Table 2** Ratios of BC/CO (mean ± standard deviation) at the five sites under different air pollution situation

| Sampling site | BC/CO ($\mu g\ m^{-3}$/ppbv) | | |
|---|---|---|---|
| | Clean | Light pollution | Heavy pollution |
| HA | $0.0048 \pm 0.008$ | $0.0057 \pm 0.013$ | – [a] |
| LH | $0.0058 \pm 0.0024$ | $0.0072 \pm 0.0023$ | $0.0071 \pm 0.0013$ |
| SX | $0.0025 \pm 0.0013$ | $0.0042 \pm 0.0013$ | – |
| WH | $0.0045 \pm 0.0022$ | $0.0042 \pm 0.0017$ | – |
| XY | $0.0048 \pm 0.0020$ | $0.0060 \pm 0.0031$ | $0.0071 \pm 0.0040$ |

[a] No heavy pollution episodes were observed



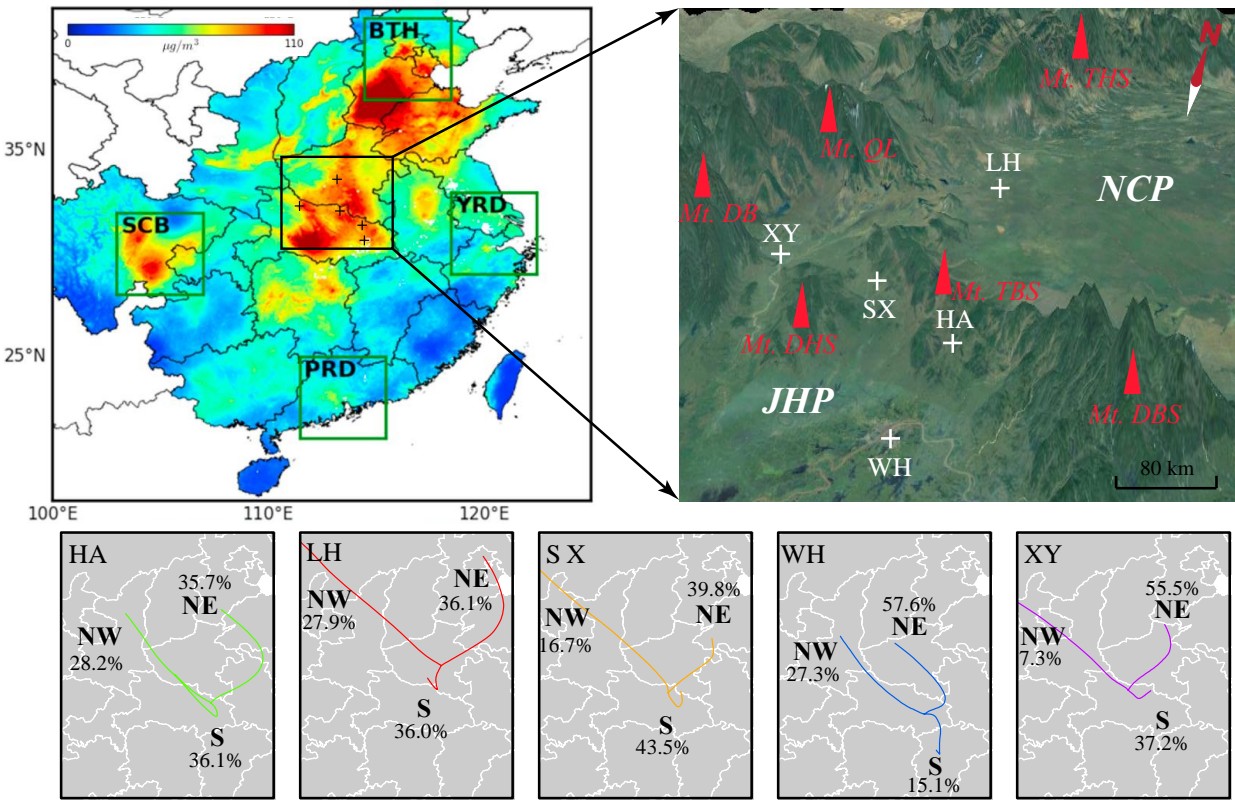

**Figure 1** Location, terrain of the study area and clusters of backward trajectories reaching at each observation site. Up left is the spatial distribution of the 15 years average of PM$_{2.5}$ concentrations at a resolution of 1 km in the study region (Lin et al., 2018). Right up shows that the study area is surrounded by mountains and Mt. DBS and Mt. TBS blocks the North China Plain (NCP) and Jianghan Plian (JHP). Bottom shows that air masses reaching at the five sites were mainly from north directions (northwest and northeast) during the observation period.



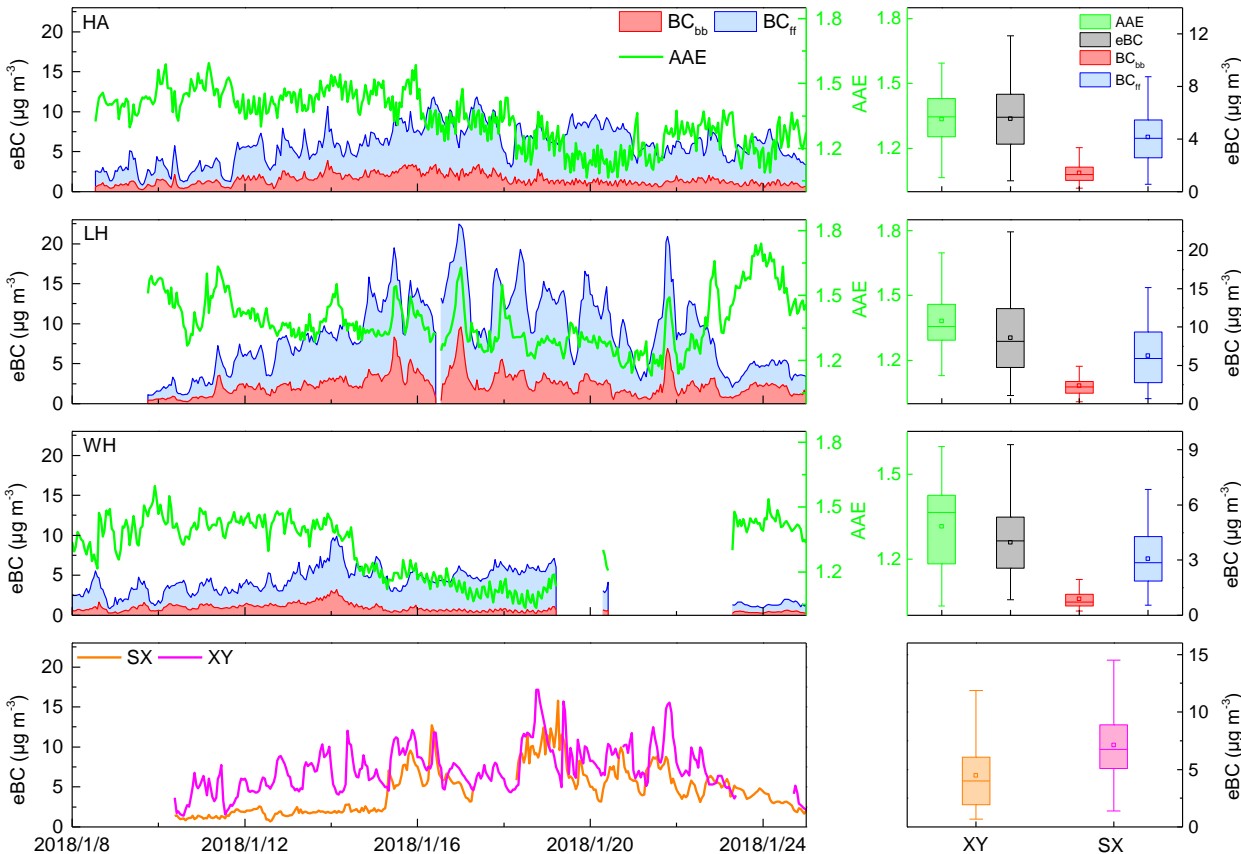

**Figure 2** Time series and box plots of eBC, BC$_{bb}$, BC$_{ff}$, and absorption Ångström exponent (AAE) at the five sites during the observation period.





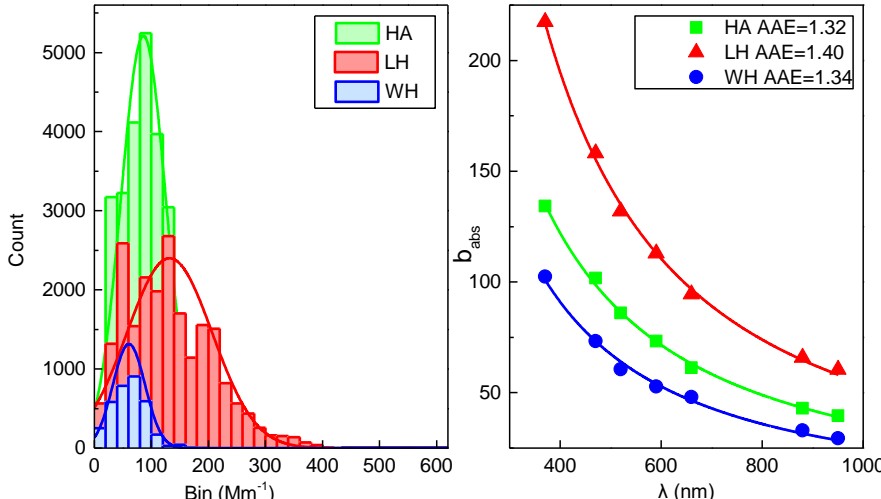

**Figure 3** Frequency distribution of absorption coefficients ($b_{abs}$) at 520 nm wavelength (left panel) and power fit of $b_{abs}$ at seven wavelengths (right panel) for HA, LH, and WH.


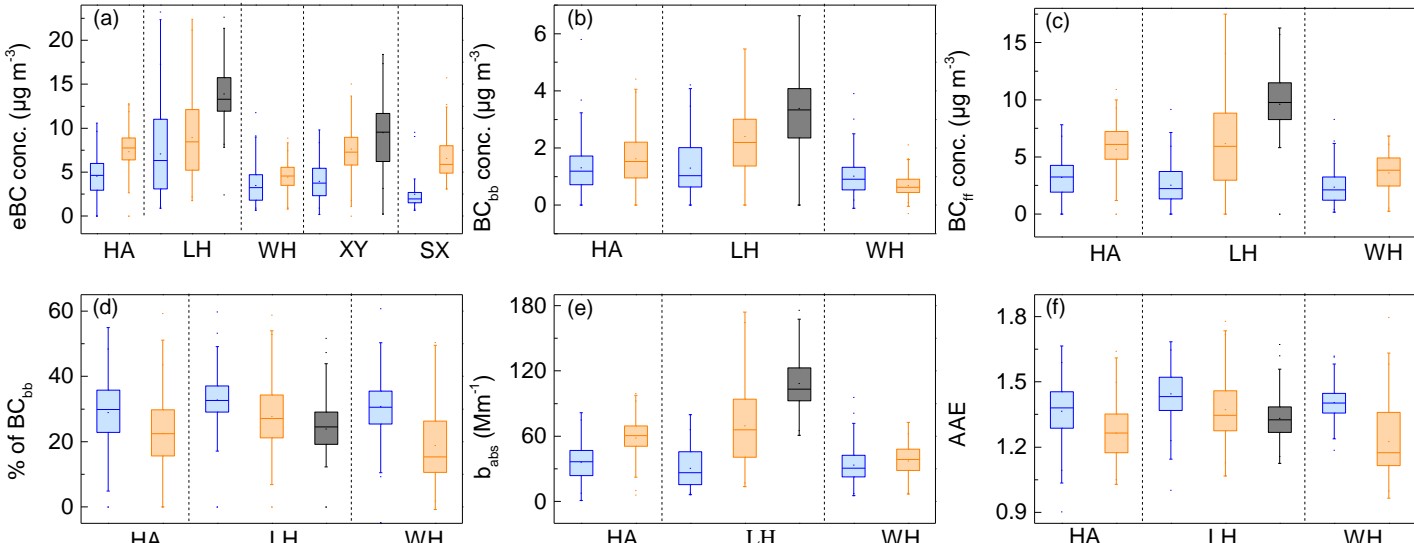

**Figure 4** Box (25−75$^{th}$ percentiles) and whisker (5−95$^{th}$ percentiles) plots of eBC concentrations (a), BC$_{bb}$ (b), BC$_{ff}$(c), percentages of BC$_{bb}$ (d), aerosol absorption coefficients (e), and absorption Ångström exponent (AAE) under different air pollution situation. The blue, orange and black color represent the clean (PM$_{2.5}$ <75 µg m$^{-3}$), light pollution (75< PM$_{2.5}$ <250 µg m$^{-3}$) and heavy pollution conditions (PM$_{2.5}$ >250 µg m$^{-3}$), respectively. The data number for the different air quality could be found in the supplementary file (Table S4).





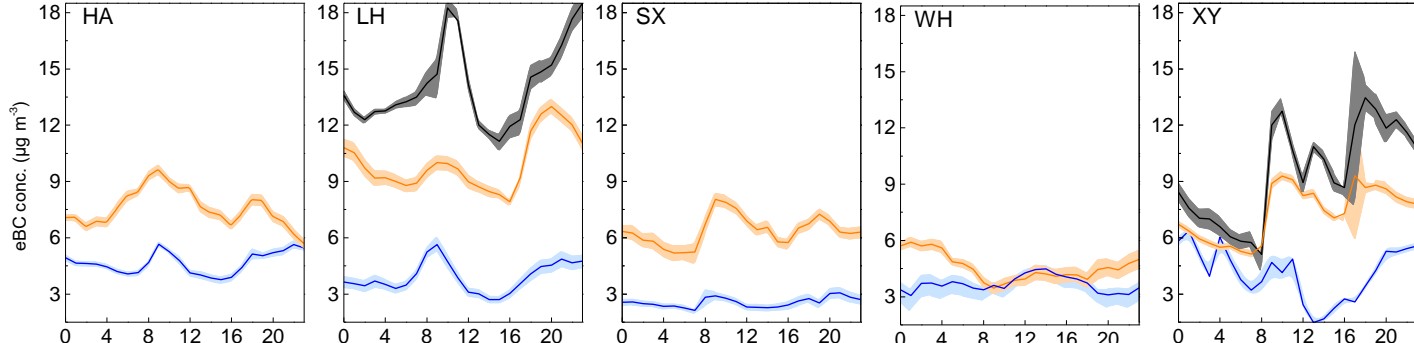

**Figure 5** Diurnal variations of eBC under different air pollution situations (blue: clean; orange: light polluted; dark: heavy polluted) at the five observation sites. The solid lines are the average values and the filled ribbons are 95[th] confidential intervals of the average value.




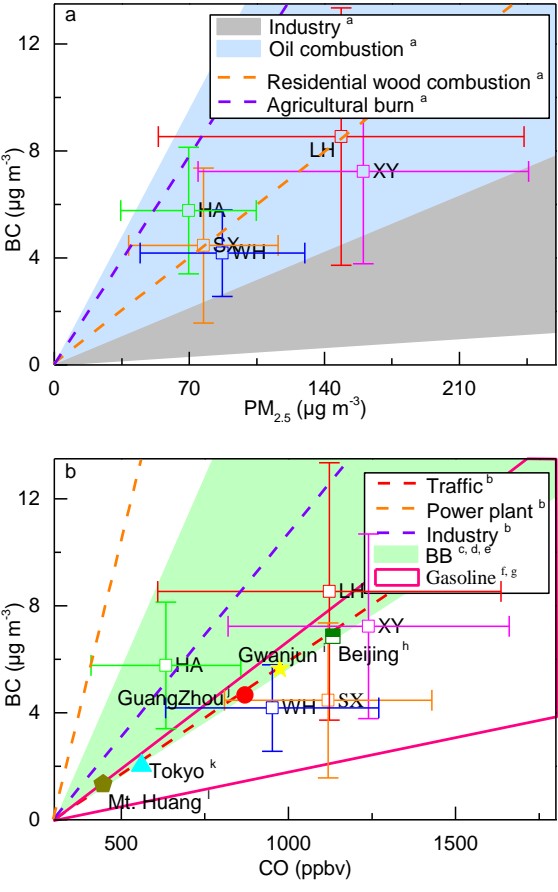

**Figure 6** Ratios of BC/PM$_{2.5}$ (a) and BC/CO (b) in this study and previous researches.

[a] Chow et al., (2011); [b] Zhang et al., (2009); [c] Dhammapala et al., (2007); [d] Cao et al., (2008); [e] Andreae and Merlet, (2001); [f] Streets et al., (2003); [g] Westerdahl et al., (2009).





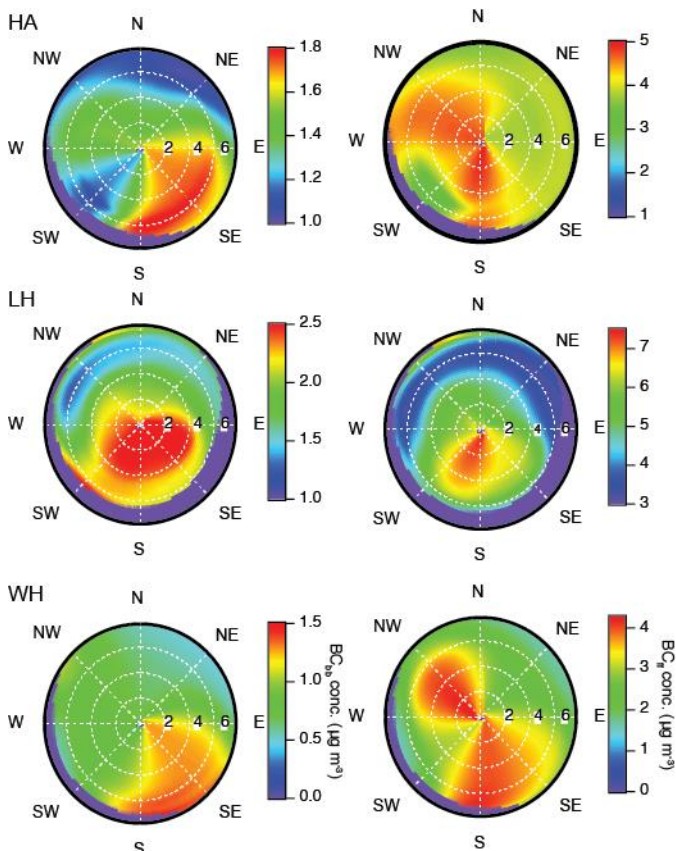

**Figure 7** NWR plots of $BC_{bb}$ (left panel) and $BC_{ff}$ (right panel) at HA, LH and WH.





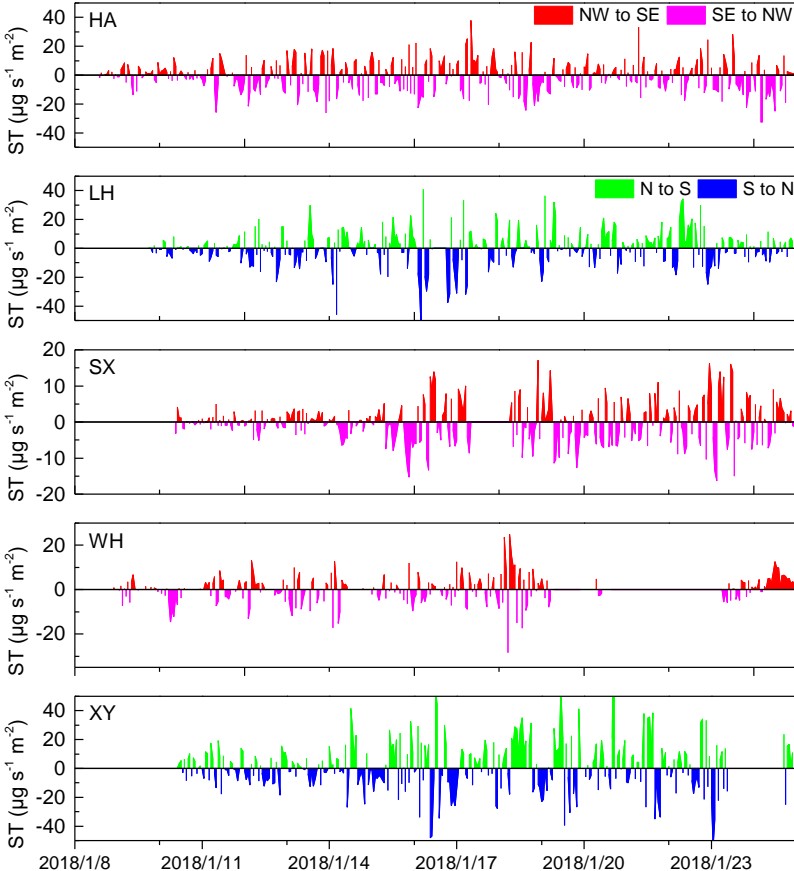

**Figure 8** Time series of surface transport intensity for BC at the five observation sites. Positive values for HA and LH indicated the transport direction was from north to south and negative values indicated the transport direction was from south to north. Positive values for SX, WH and XY indicated the transport directions were from northwest to southeast and negative values indicated the transport directions were from southeast to northwest.



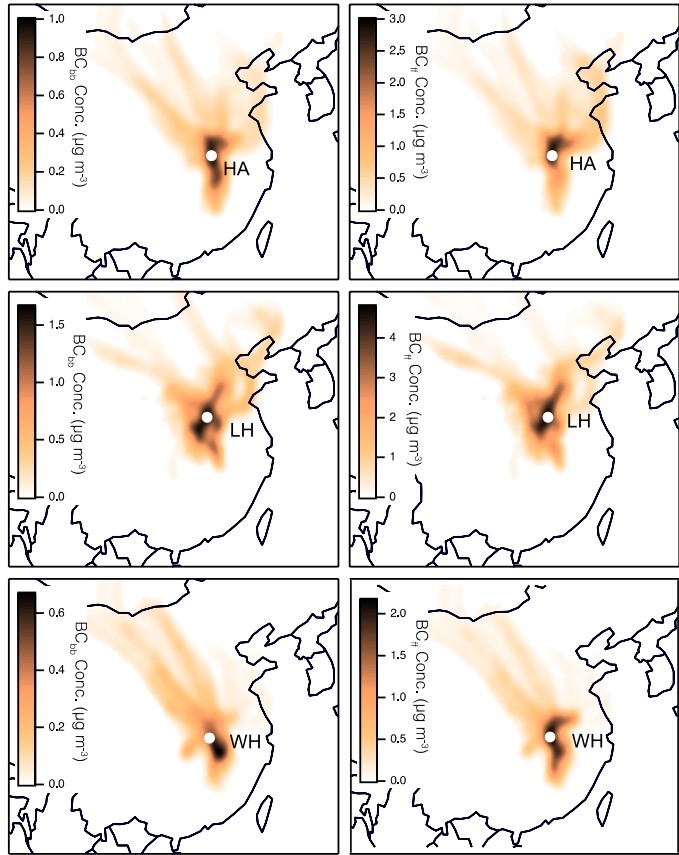

**Figure 9** Concentration-weighted trajectory (CWT) plots of BC$_{bb}$ (left panel) and BC$_{ff}$ (right panel) at HA, LH and WH during the whole observation site. The white dote represents the observation site.



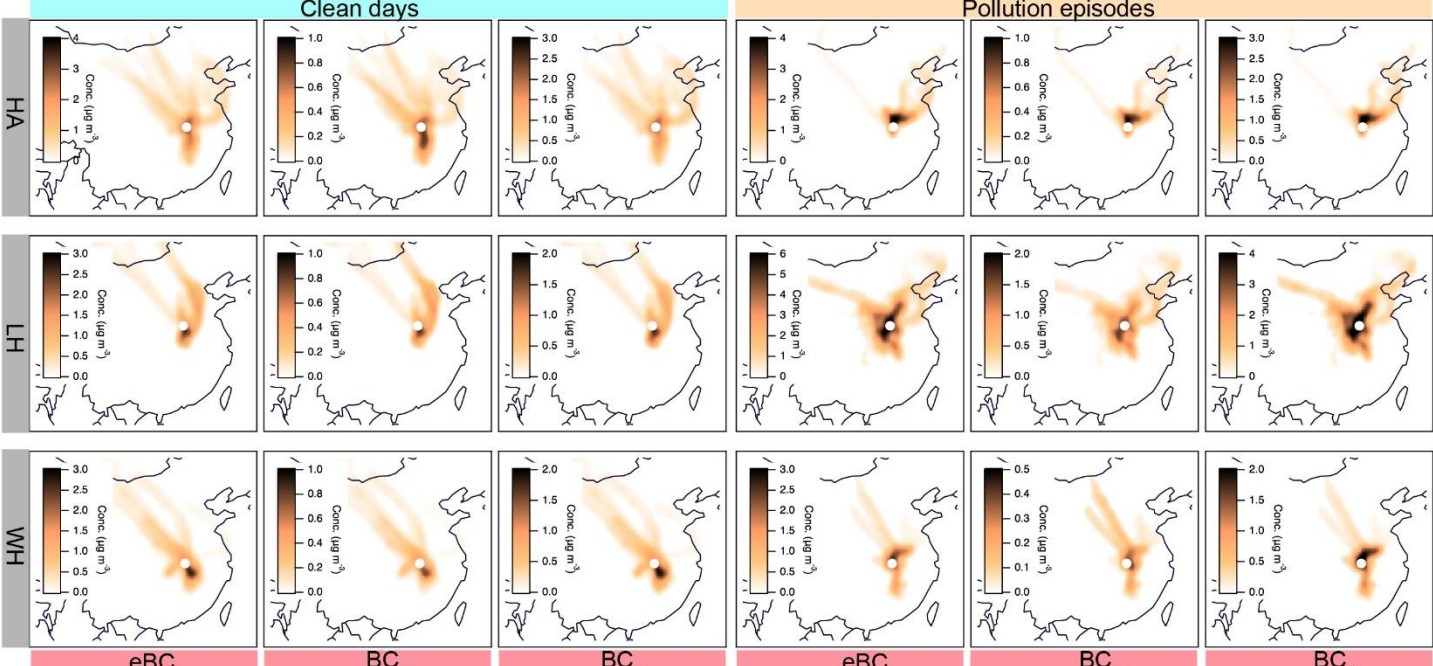

**Figure 10** Concentration-weighted trajectory (CWT) plots of eBC, $BC_{bb}$ and $BC_{ff}$ during clean and pollution episodes at HA, LH and WH. The white dot represents the observation site.





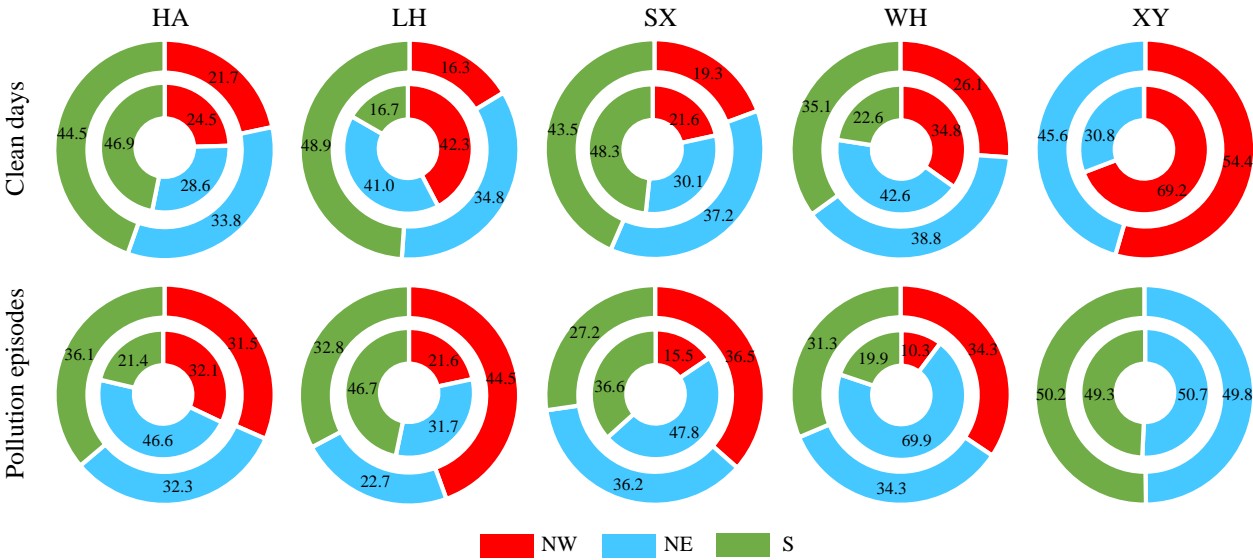

**Figure 11** Cluster results at five sites (inner pie plots) and the eBC percentage contributions from different clusters (extern pie plot) during the clean days (up panel) and pollution episodes (bottom panel). NW, NE and S mean the northwest, northeast and south direction clusters as shown in Figure 1.





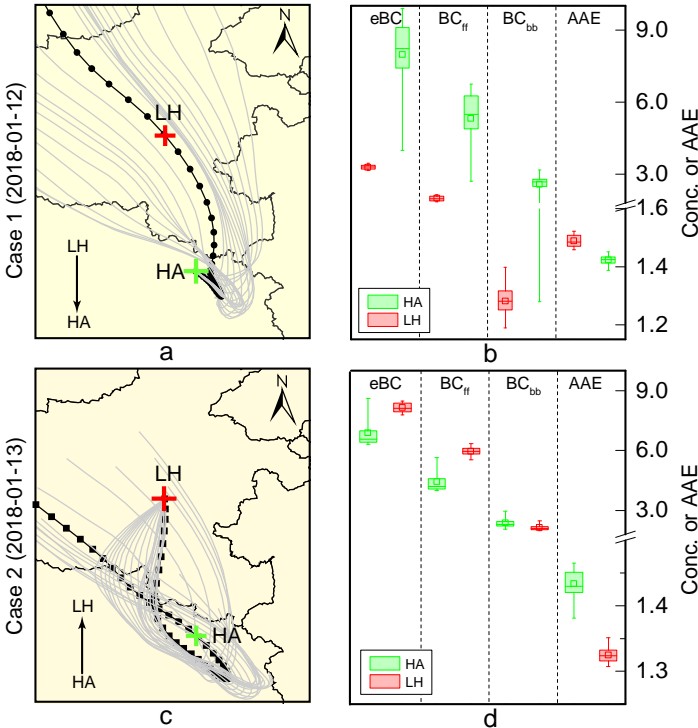

**Figure 12** Case study of BC variation during the transportation from upwind to downwind direction. a (case 1): Hourly backward trajectories (grey line) reaching at HA on 2018-1-12 and the trajectory at 13:00 (UTC) (black line) was found passing through LH about 28 hours ago. c (case 2): Trajectory reaching at LH on 2018-1-13 07:00 (UCT) (black line) was found passing through HA about 31 hours ago. Box (25-75[th] percentiles) and whisker (5-95[th] percentiles) plots of eBC, BC$_{ff}$, BC$_{bb}$ and AAE variation during the transport from LH to HA (b) and from HA to LH (d).