# Peer review of "Intra-regional transport of black carbon between the south edge of"

_Atmospheric Chemistry and Physics, 2018_

## Referee Comment (RC1) · Anonymous Referee #1 · 13 Dec 2018

The authors present a simultaneous field measurement dataset of BC at five sites in this paper with the aim to investigate the intra-regional transport between the south edge of North China Plain and Central China based on the variations of BC mass concentration, sources and optical properties. The dataset is important, and would be with good scientific significance to help people to model the BC aerosol climate effect in East Asian by studying the changes of BC physio-chemical and optical properties during the transport. My major concern is, as the authors stated in their paper (in the introduction), one of the key purpose of this study were to quantify the regional transportation of BC at multiple observation sites in CC and SE-NCP.... But the backward trajectory method used by the authors can just get some qualitative analysis of the

air parcels transport as presented in their study. The paper will be greatly improved if the authors consider using models to simulate the emissions and then to quantify the intra-regional contributions at the sites based on the measured BC concentration data. In addition, there are also a lot of language issues and editing needs that have to be addressed. The authors thus need to make a careful revision and correction on the language, especially revisions on some seemingly illogical expression, to improve the overall quality of the paper for publication in the journal. I would recommend the editor to reconsider the papers after a major revision by the authors.

Other specific comments,

Section 3.1, the authors should more focus on discussing and comparing the different BC levels between the studied 5 sites and other regions in China or over the globe, not on North China and other regions.

Line 214, "...Despite the sampling periods, site types, inlet of aerosol and 213 instruments were different between different studies (Table S1), BC was generally higher in North China and lower BC levels were found in remote areas and coastal areas... " Here, it is not appropriate and logical expression by saying the two things using the "despite".

Line 242 "..At WH, the concentration and percentage of BCbb both decreased from clean to pollution, which suggested that more BCff was emitted during haze episodes...", should revise as "... both the concentration and percentage of BCbb decreased from clean to pollution"..." also, you say more fossil fuel BC was emitted. But the increased BC is probably due to the accumulation of pollutes during polluted days when the PBL is lowered.

Lines 248-250, "...the pollution episodes (Huang et al., 249 2014), and the increased secondary aerosols would be more adsorbed on the surface of BC...." Do you mean that more secondary aerosols will be coated on BC? "...the $\sigma$abs also elevated by 11.7$-$254% as the air quality switched from clean to pollution (Fig. 4e). There are

more secondary aerosols (i.e., sulfate, nitrate) during the pollution episodes (Huang et al., 249 2014),.." Do you have some observations of chemical composition that would support your conclusions.

Line 253, "The decreasing of AAE from clean to polluted days was also reported elsewhere (Zhang et al., 2015b) and it can be partly attributed to the source variation...", the AAE is very sensitive to particles size, so you may need to think about the particles growth due to the secondary formation processes.

Lines 257-267, you only show the diurnal variations of mass concentrations of BC, how about the absorption coefficient?

Line 271, You say "...combustion (traffic) and agricultural burning are higher than those from industrial emissions such as manufacturing and mineral products. But you give an example for the lower ratios from residential wood combustion, which is not an industrial source.

L291, It would be more interesting if you discuss whether the BC at downwind sites is more aged because of the transportation, because you say that "...the BC/CO is used to reflect the BC aging during the transport.

Line 307, "…...The same result was also found at WH. High level of BCbb was due to more biomass burning in the southeast direction of HA and WH..."How do you know that more biomass burning in the southeast? Do you have some evidences to support your statement?

Section 3.5, The paragraph may need to revise very carefully for that the current statements on the influences of the air parcels (CWT analysis) on each sites are too trivial and wordy to understand. The authors are suggested to simulate the emissions and to quantify the intra-regional contributions at the sites based on the measured BC concentration data by using regional models.

Line 367, "...the travelling time (aging time) from LH to HA and from HA to LH were 28

h and 31 h, respectively, which suggested that the BC particle should be coagulated through complex atmospheric processes. Therefore, the new emission inputs along the trajectory enhanced the eBC mass concentration during the transport. . ..” How do you infer that “the new emission” enhanced the eBC mass concentration from the previous sentence (longer aging time and coagulation processes) here?

Line 370, “. . .However, slight differences found for BCbb transport: BCbb increased from LH (1.28 ± 0.06 $\mu$g m−3) to HA (2.57 ± 0.47 $\mu$g m−3), while BCbb decreased from HA (2.37 ± 0.23 $\mu$g m−3) to LH (2.14 ± 0.14 $\mu$g m−3). . .” What do you mean about this?

---

## Referee Comment (RC2) · Anonymous Referee #2 · 6 Jan 2019

This study investigates the intra-regional transportation of black carbon (BC) between North China plain (NCP) and central China (CC) based on the simultaneously measurements at five cities located in the two regions during winter haze period. The authors have identified two important BC emission sources (i.e., biomass burning and fossil fuel) and their geographic origins during transportation. Since there are still limited studies on the intra-region transportation in China, this study takes insight in this topic. The manuscript is well written and organized. But there are still some minor issues, which need to be addressed before publication. Please see specific comments below. 1. Lines 240-246: Taking the fossil fuel BC (BCff) as the control priority in WH and other cities in this study is not just because the BCff increases from clean to

pollution period but also due to the much higher absolute conc. of BCff than biomass burning BC (BCbb) under all the three conditions (Fig. 4b and c). Regarding Beijing case, in addition to the percentage of BBbb increasing from clean to pollution episodes, are the absolute conc. of BCbb also higher than BCff? If not, it should be careful to state that the priority in North China is to control BCbb. 2. The aging process could significantly change the optical properties of BC aerosols (Peng J, et al., PNAS, 2016; Wang et al., J. Adv. Model Earth Syst, 2018). Are there any observed changes in BC absorption due to the aging during the intra-regional transportation?

Figures: Fig.1: For air mass clustering panels, you may want to use different colors to differentiate the air masses from different directions. Fig. 3: Does the count (y axis) denote the number of data points? If so, why there is much less data points at WH? The smaller total count might be because missing or not available data in measurements at WH? Typos Line 227: Figure 3 is Figure 3b.

---

## Author Comment (AC2) · 16 Feb 2019

This study investigates the intra-regional transportation of black carbon (BC) between North China plain (NCP) and central China (CC) based on the simultaneously measurements at five cities located in the two regions during winter haze period. The authors have identified two important BC emission sources (i.e., biomass burning and fossil fuel) and their geographic origins during transportation. Since there are still limited studies on the intra-region transportation in China, this study takes insight in this topic. The manuscript is well written and organized. But there are still some minor issues, which need to be addressed before publication. Please see specific comments below.

AR: Thanks for your positive comments on this manuscript.

1. Lines 240-246: Taking the fossil fuel BC ($BC_{ff}$) as the control priority in WH and other cities in this study is not just because the $BC_{ff}$ increases from clean to pollution period but also due to the much higher absolute conc. of $BC_{ff}$ than biomass burning BC ($BC_{bb}$) under all the three conditions (Fig. 4b and c). Regarding Beijing case, in addition to the percentage of $BC_{bb}$ increasing from clean to pollution episodes, are the absolute conc. of BCbb also higher than $BC_{ff}$? If not, it should be careful to state that the priority in North China is to control BCbb.

AR: Thanks for your comments and we have revised corresponding part. You concerns whether the absolute concentration of $BC_{bb}$ was higher than $BC_{ff}$ during pollution episodes in Beijing (Liu et al., 2018), the answer is yes as shown in Fig. R6. BC source apportionment using carbon isotope also suggests that BC emissions in BTH (North China) and PRD (South China) are characterized by coal-combustion-dominated and liquid fossil-combustion-dominated, respectively (Yu et al., 2018). This result supported our conclusion.

[Figure]

Figure. R6 Liquid and solid fuel source contributions to BC in pollution episodes (EP) and clean days (CD). The size of pies is proportional to average concentration of BC in each episode (Liu et al., 2018).

References

Liu, Y., Yan, C. and Zheng, M.: Source apportionment of black carbon during winter in Beijing, Science of The Total Environment, 618, 531–541, doi:10.1016/j.scitotenv.2017.11.053, 2018.

Yu, K., Xing, Z., Huang, X., Deng, J., Andersson, A., Fang, W., Gustafsson, Ö., Zhou, J. and Du, K.: Characterizing and sourcing ambient $PM_{2.5}$ over key emission regions in China III: Carbon isotope based source apportionment of black carbon, Atmospheric Environment, 177, 12–17, doi:10.1016/j.atmosenv.2018.01.009, 2018.

2. The aging process could significantly change the optical properties of BC aerosols (Peng J, et al., PNAS, 2016; Wang et al., J. Adv. Model Earth Syst, 2018). Are there any observed changes in BC absorption due to the aging during the intra-regional transportation?

AR: Thanks for your suggestion. We have observed the enhancement of BC absorption during the transportation according to two cases as shown in Fig. R7. In case 1, $\sigma_{abs}$ significantly (p<0.01) increased from 25.6 ± 0.81 Mm$^{-1}$ (LH) to 61.8 ± 12.5 Mm$^{-1}$ (HA). In case 2, the enhancement of $\sigma_{abs}$ was also observed from HA (53.4 ± 5.58 Mm$^{-1}$) to LH (59.9 ± 2.05 Mm$^{-1}$).

[Figure]

Figure. R7 Case studies of BC variation during the transportation from upwind to downwind direction. a (case 1): Hourly backward trajectories (grey line) reaching at HA on 2018-1-12 and the trajectory at 13:00 (GMT) (black line) was found passing through LH about 28 hours ago. c (case 2): Trajectory reaching at LH on 2018-1-13 07:00 (GMT) (black line) was found passing through HA about 31 hours ago. Box (25-75[th] percentiles) and whisker (5-95[th] percentiles) plots of eBC, BC$_{ff}$, BC$_{bb}$ $\sigma_{abs}$, and AAE variations during the transport from LH to HA (b) and from HA to LH (d).

3. Fig.1: For air mass clustering panels, you may want to use different colors to differentiate the air masses from different directions.

AR: Thanks for your suggestions and we have corrected it (Fig. R8).

[Figure]

Figure. R8 Location, terrain of the study area and clusters of backward trajectories reaching at each observation site. Up left is the spatial distribution of the 15 years average PM$_{2.5}$ concentrations at a resolution of 1 km (Lin et al., 2018). Right up shows that the study area is surrounded by mountains and Mt. DBS and Mt. TBS blocks the North China Plain (NCP) and Jianghan Plian (JHP). Bottom shows that air masses reaching at the five sites were mainly from north directions (northwest and northeast) during the observation period.

4. Fig. 3: Does the count (y axis) denote the number of data points? If so, why there is much less data points at WH? The smaller total count might be because missing or not available data in measurements at WH?

AR: Yes, the y axis represents the number of data, the reasons why the number of data in WH ls less than other sites is due to the following two reasons: (1) there were much more missing values due to the instrument maintaining and power failure (2) the data resolution is 5-min at WH (AE31) and the resolution is 1-min at LH and HA (AE33).

5. Typos Line 227: Figure 3 is Figure 3b.

AR: Thanks, and it has been corrected.

---

## Author Response (AR1)

The authors present a simultaneous field measurement dataset of BC at five sites in this paper with the aim to investigate the intra-regional transport between the south edge of North China Plain and Central China based on the variations of BC mass concentration, sources and optical properties. The dataset is important and would be with good scientific significance to help people to model the BC aerosol climate effect in East Asian by studying the changes of BC physio-chemical and optical properties during the transport. My major concern is, as the authors stated in their paper (in the introduction), one of the key purpose of this study were to quantify the regional transportation of BC at multiple observation sites in CC and SE-NCP. But the backward trajectory method used by the authors can just get some qualitative analysis of the air parcels transport as presented in their study. The paper will be greatly improved if the authors consider using models to simulate the emissions and then to quantify the intra-regional contributions at the sites based on the measured BC concentration data. In addition, there are also a lot of language issues and editing needs that have to be addressed. The authors thus need to make a careful revision and correction on the language, especially revisions on some seemingly illogical expression, to improve the overall quality of the paper for publication in the journal. I would recommend the editor to reconsider the papers after a major revision by the authors.

AR: Thanks for your comment. We have added the Geos-Chem simulation to quantify the transport contributions. However, the Geos-Chem results are not good enough and details are provided in the response to Comment 10. Additionally, we have also carefully checked and polished the language.

Other specific comments,

1. Section 3.1, the authors should more focus on discussing and comparing the different BC levels between the studied 5 sites and other regions in China or over the globe, not on North China and other regions.

AR: Thanks for your suggestion and we do the literatures review. Comparison of BC in this study and other regions was listed in the supplementary file Fig. R1:

[Figure]

Figure. R1 Spatial distribution of BC mass concentration (a) and absorption coefficients (b) in China. More details can be found in Table S1 and S2 in the supplementary materials.

*As Table S1 shown, BC was generally higher in North China and lower BC levels were found in remote areas and coastal areas as Fig. 3a shows. Wang et al, (2014b) analyzed ambient BC in an urban site in Xi'an during winter and found the average mass concentration was $8.8 \pm 3.7 \ \mu g \ m^{-3}$, which was higher than that in this study. Compared to other regions, BC levels in this study were higher than a remote area of Lulang in southeastern part of the Tibetan Plateau (0.31*

*± 0.55 µg m⁻³) (Wang et al., 2018) as well as coastal areas such as Hong Kong (1.4 ± 1.1 µg m⁻³) (Wang et al., 2017a) and a rural site in Shenzhen (2.6 ± 1.0 µg m⁻³) (Huang et al., 2012).*

References

Huang, X.F., Sun, T.L., Zeng, L.W., Yu, G.H. and Luan, S.J.: Black carbon aerosol characterization in a coastal city in South China using a single particle soot photometer, Atmos. Environ., 51, 21–28, doi:10.1016/j.atmosenv.2012.01.056, 2012.

Wang, Q., Huang, R.J., Cao, J., Han, Y., Wang, G., Li, G., Wang, Y., Dai, W., Zhang, R. and Zhou, Y.: Mixing state of black carbon aerosol in a heavily Polluted urban area of China: Implications for light Absorption enhancement, Aerosol Sci. Technol., 48(7), 689–697, doi:10.1080/02786826.2014.917758, 2014b.

Wang, J., Virkkula, A., Gao, Y., Lee, S., Shen, Y., Chi, X., Nie, W., Liu, Q., Xu, Z., Huang, X., Wang, T., Cui, L. and Ding, A.: Observations of aerosol optical properties at a coastal site in Hong Kong, South China, Atmos. Chem. Phys., 17(4), 2653–2671, doi:10.5194/acp-17-2653-2017, 2017a.

Wang, Q., Cao, J., Han, Y., Tian, J., Zhu, C., Zhang, Y., Zhang, N., Shen, Z., Ni, H., Zhao, S. and Wu, J.: Sources and physicochemical characteristics of black carbon aerosol from the southeastern Tibetan Plateau: internal mixing enhances light absorption, Atmos. Chem. Phys., 18(7), 4639–4656, doi:10.5194/acp-18-4639-2018, 2018.

2. Line 214, ". . .Despite the sampling periods, site types, inlet of aerosol and instruments were different between different studies (Table S1), BC was generally higher in North China and lower BC levels were found in remote areas and coastal areas... " Here, it is not appropriate and logical expression by saying the two things using the "despite".

AR: We have corrected it as above.

3. Line 242 "..At WH, the concentration and percentage of $BC_{bb}$ both decreased from clean to pollution, which suggested that more BCff was emitted during haze episodes...", should revise as "... both the concentration and percentage of $BC_{bb}$ decreased from clean to pollution". . ." also, you say more fossil fuel BC was emitted. But the increased BC is probably due to the accumulation of pollutes during polluted days when the PBL is lowered.

AR: Thanks for your correction, and we have revised it.

4. Lines 248-250, ". . .the pollution episodes (Huang et al., 249 2014), and the increased secondary aerosols would be more adsorbed on the surface of BC. . .." Do you mean that more secondary aerosols will be coated on BC? "...the σabs also elevated by 11.7−254% as the air quality switched from clean to pollution (Fig. 4e). There are more secondary aerosols (i.e., sulfate, nitrate) during the pollution episodes (Huang et al., 249 2014),.." Do you have some observations of chemical composition that would support your conclusions.

AR: Yes, we have conducted the off-line low volume $PM_{1.0}$ sampling and chemical analysis at the five sites during the field sampling campaign. Here, we just show part of the chemical analysis results (which have not been published) to support that there were more sulfate and nitrate during the pollution episodes as shown in the following figure.

[Figure]

Fig. R2 Daily concentrations of sulfate (green line) and nitrate (blue line) during observation (left panel) and their average concentrations for clean days (blue) and pollution episodes (orange) (right panel).

5. Line 253, "The decreasing of AAE from clean to polluted days was also reported else- where (Zhang et al., 2015b) and it can be partly attributed to the source variation. . .", the AAE is very sensitive to particles size, so you may need to think about the particles growth due to the secondary formation processes.

AR: Thanks for your comments and we agree with you. However, we did mot measure the particle size during our observation, and we try our best to explain the decreasing of AAE during pollution episodes according to previous studies and the sentences are revised as following:

*The AAE is also sensitive to other factors such as the particle size. Previous studies suggested that the particle diameter and number concentration increased from clean to pollution episodes due to several factors such as coagulation, hygroscopic growth, emissions, meteorological conditions, i.e., planetary boundary layer and wind speed (Guo et al., 2014; Zhang et al., 2017). These studies suggested that the particle diameter is generally larger during pollution days. Furthermore, the lab combustion and numeric simulation proved that BC particle with larger geometric median diameter had lower AAE value (Singh et al., 2016; Liu et al., 2018b). Therefore, lower AAE was observed during pollution episodes in this study.*

References

Guo, S., Hu, M., Zamora, M. L., Peng, J., Shang, D., Zheng, J., Du, Z., Wu, Z., Shao, M., Zeng, L., Molina, M. J. and Zhang, R.: Elucidating severe urban haze formation in China, Proceedings of the National Academy of Sciences, 111(49), 17373–17378, doi:10.1073/pnas.1419604111, 2014.

Liu, C., Chung, C. E., Yin, Y. and Schnaiter, M.: The absorption Ångström exponent of black carbon: from numerical aspects, Atmospheric Chemistry and Physics, 18(9), 6259–6273, doi:10.5194/acp-18-6259-2018, 2018.

Singh, S., Fiddler, M. N. and Bililign, S.: Measurement of size-dependent single scattering albedo of fresh biomass burning aerosols using the extinction-minus-scattering technique with a combination of cavity ring-down spectroscopy and nephelometry, Atmospheric Chemistry and Physics, 16(21), 13491–13507, doi:10.5194/acp-16-13491-2016, 2016.

Zhang, K., Wang, D., Bian, Q., Duan, Y., Zhao, M., Fei, D., Xiu, G. and Fu, Q.: Tethered balloon-based particle number concentration, and size distribution vertical profiles within the lower troposphere of Shanghai, Atmospheric Environment, 154, 141–150, doi:10.1016/j.atmosenv.2017.01.025, 2017.

6. Lines 257-267, you only show the diurnal variations of mass concentrations of BC, how about the absorption coefficient?

AR: Thanks for your suggestion, and we found that the diurnal variations of absorption coefficient and BC levels were similar as the following figure shown.

[Figure]

Fig. R3 Diurnal variations of BC absorption coefficients. (The figure has been added as Fig. S7 in the supplementary materials of the revised version)

In the revised manuscript, we have revised this part as following:

*Figure 6 and Fig. S7 shows the diurnal variations of eBC and absorption coefficients under different air quality. The diurnal cycles of black carbon and absorption showed similar variation patterns. The BC mass concentrations were discussed here.*

7. Line 271, You say ". . .combustion (traffic) and agricultural burning are higher than those from industrial emissions such as manufacturing and mineral products. But you give an example for the lower ratios from residential wood combustion, which is not an industrial source.

AR: Thanks for your comment. We have carefully checked the reference (Chow et al., 2011). The ratios of BC/PM$_{2.5}$ from mobile sources and area sources were generally higher than that from industrial sources and we have revised this part as following:

*Generally, the ratios of BC/PM$_{2.5}$ from mobile sources (0.059-0.74) and area sources (0.032-0.33) were higher than that from industrial sources (0.0046-0.03). For instance, the mobile sources hold the highest ratios of BC/PM$_{2.5}$ (0.33−0.77) and the cement kiln showed lower ratio (0.03) (Chow et al., 2011).*

References

Chow, J. C., Watson, J. G., Lowenthal, D. H., Antony Chen, L.-W. and Motallebi, N.: PM$_{2.5}$ source profiles for black and organic carbon emission inventories, Atmos. Environ.,, 45(31), 5407–5414, doi:10.1016/j.atmosenv.2011.07.011, 2011.

8. L291, It would be more interesting if you discuss whether the BC at downwind sites is more aged because of the transportation, because you say that ". . .the BC/CO is used to reflect the BC aging during the transport.

AR: We feel sorry that we did not discuss the aging of BC during the transport by BC/CO ratio. We have tried to discuss the BC/CO ratio using the same method in section 3.6, however, the low data resolution of CO (1-hour) would cause large uncertainty. So, we used the AAE instead of BC/CO ratio to discuss the BC aging during the transportation from upwind to downwind site. The decreasing of AAE from upwind to downwind site suggested that the BC was aged during the transportation and more details can be found in section 3.6.

9. Line 307, ". . ...The same result was also found at WH. High level of BCbb was due to more biomass burning in the southeast direction of HA and WH. . ."How do you know that more biomass burning in the southeast? Do you have some evidences to support your statement?

AR: We draw this conclusion because there were more fire spots in the southeast direction of WH during the study period as shown in Fig. R4. We have revised this part.

[Figure]

Figure R4 Locations of the fire spots downloaded from MODIS during the observation period (2018/1/08~2018/1/25).

10. Section 3.5, The paragraph may need to revise very carefully for that the current statements on the influences of the air parcels (CWT analysis) on each site are too trivial and wordy to understand. The authors are suggested to simulate the emissions and to quantify the intra-regional contributions at the sites based on the measured BC concentration data by using regional models.

AR: Thanks for your suggestion, we have tried Geos-Chem model to simulate the intra-regional transport contribution. The settings of Geos-Chem are described as the following part:

We use the nested GEOS-Chem model for China (version 11-01, http://wiki.seas.harvard.edu/geos-chem/index.php/Main_Page) to simulate the surface BC concentration. Driven by the GEOS-FP assimilation meteorology from the Goddard Earth Observing System (GEOS) of the NASA Global Modeling and Assimilation Office, the nested model has a horizontal resolution of 0.3125° longitude × 0.25° latitude with 47 vertical layers, and the lowest 10 layers are of ~130 m thickness each. The lateral boundary conditions of nested model are taken every 3 hours from a global GEOS-Chem simulation at 2.5° long × 2° lat horizontally. Spin-up time for nested model and global model are 15 days and one month, respectively. The scheme of planetary boundary layer employs a non-local scheme following Lin et al. (2010). Model convection is simulated with the relaxed Arakawa–Schubert scheme (Rienecker et al., 2008). Both the global and nested GEOS-Chem models are run with the $NO_x$-$O_x$-hydrocarbon-aerosol-bromine tropospheric chemistry mechanism with online aerosols. Aerosols simulated by model include secondary inorganic aerosols (SIOA, including sulfate, nitrate and ammonium), secondary organic aerosols (SOA), primary organic aerosols (POA), black carbon (BC), dust and sea salts.

Monthly gridded anthropogenic emissions in China are taken from the Multi-resolution Emission Inventory for China (MEIC, www.meicmodel.org; Geng et al., 2017) of 2016 for nitrogen oxides ($NO_x$), carbon monoxide (CO), sulfur dioxide ($SO_2$), BC and POA. Following Zhang et al. (2015), emissions of anthropogenic fine dust are also included as primary $PM_{2.5}$ excluding BC and POA from MEIC in 2012. Biomass burning emissions are taken from the monthly GFED4 datasets (Giglio et al., 2013). Biogenic emissions of NMVOC follow MEGANv2.1 (Guenther et al., 2012). Soil emissions of $NO_x$ employ the parameterization from Hudman et al. (2012).

Control and two sensitivity simulations were also done to study the regional transport contribution. The settings of control simulation were described above, and the sensitivity simulation were done with the emissions from Hubei and Henan province being closed, respectively.

The simulated and observed time series of BC at the five sites are shown in the Fig. R5. The Pearson coefficients (*r*) and NMB ranged from -0.44 to 0.07 and -39.9 to 19.8%, respectively, which suggested that the Geos-Chem is not good enough to reconstruct the BC variation in this study. There are several reasons: (1) the emission inventory uncertainty due to activity data, emission factors for energy-related combustion; burned area, fuel load and combustion completeness and emission factor for open burning emissions (Bond et al., 2013); low temporal resolution (i.e., monthly in this study); (2) the uncertainty of the input reanalysis meteorological field (i.e., in this study); (3) simple physical-chemical mechanism of BC in code, etc. More accurate and quantitative modeling for regional transportation of BC should be done after the above problems improved in the future.

Considering the poor simulation result, the Geos-Chem results were not adopted in the revised manuscript. Additionally, we carefully revised section 3.5 as the following:

*Employing CWT method, the potential geographic origins of eBC for the five sites were explored (Fig. S11). Overall, CWT results of eBC at the five sites suggested that high eBC levels were found both in the north and south directions of LH and WH, while the high levels (i.e., > 4 μg m$^{-3}$) of eBC were only found from northeast directions of HA, SX and XY (Fig. S11). Additionally, the potential geographic source regions of BC$_{bb}$ and BC$_{ff}$ at HA, LH and WH were also discussed as shown in Fig. 10. At HA, the CWT results showed that high levels of eBC (i.e., > 3 μg m$^{-3}$) were from north/northeast direction. However, the hot spots of BC$_{bb}$ and BC$_{ff}$ were different, with higher levels of BC$_{bb}$ from both south and north*

*directions and higher levels of BC$_{ff}$ from the north direction. Also, higher levels of BC$_{bb}$ and BC$_{ff}$ were found in the south of LH. Opposite to the CWT results at HA, the hot spots of BC$_{bb}$ was only found in the southeast direction of WH and high levels of BC$_{ff}$ were found in the north and south directions of WH. The CWT results at WH were in line with the CBPF plots in section 3.4. The unity of CWT and CBPF results at WH suggested that there were intensive biomass burning activities in the south direction of WH during the observation period, which was verified by the MODIS fire-points distribution (Fig. S10).*

*We also discussed the source region differences of BC under different air quality (Fig. 11). The higher levels (>1 μg m$^{-3}$) of eBC, BC$_{bb}$ and BC$_{ff}$ were mainly from the south direction of three sites when the air was clean, while during the pollution episodes, air parcels from the north direction contributed high concentrations. For instance, at WH, high levels of eBC (> 2.5 μg m$^{-3}$) were found from south direction, while the source regions with high level eBC (> 3 μg m$^{-3}$) switched to northeast direction when the air quality was worsened. Figure 12 shows the semiquantitative results of transportation contribution results during clean and pollution episodes. At the boundary sites (HA, SX and XY), BC was mainly from south direction (accounting for 46.0−58.2%) when the air quality was clean, and it was mainly from northeast/northwest directions (51.2−76.5%) when the air quality getting worse. At SE-NCP site (LH), BC was dominantly from south direction (47.8%) during pollution episodes. At CC site (WH), BC was mainly from northeast direction (49.3−71.1%). These results suggested that northwest and northeast directions were the main transport pathways of air pollutants reaching to WH during the pollution episodes. Furthermore, to control local emissions during haze episodes, the emission sources, i.e., industry plant and open biomass burning in the upwind direction should also be controlled to prevent the further deterioration of air quality in downwind areas.*

[Figure]

Figure. R5 Time series of observed and simulated BC concentrations from Goes-Chem model during the study period.

References

Bond, T. C., Doherty, S. J., Fahey, D. W., Forster, P. M., Berntsen, T., DeAngelo, B. J., Flanner, M. G., Ghan, S., Kärcher,

B., Koch, D., Kinne, S., Kondo, Y., Quinn, P. K., Sarofim, M. C., Schultz, M. G., Schulz, M., Venkataraman, C., Zhang, H., Zhang, S., Bellouin, N., Guttikunda, S. K., Hopke, P. K., Jacobson, M. Z., Kaiser, J. W., Klimont, Z., Lohmann, U., Schwarz, J. P., Shindell, D., Storelvmo, T., Warren, S. G. and Zender, C. S.: Bounding the role of black carbon in the climate system: A scientific assessment, J. Geophys. Res, Atmos., 118(11), 5380–5552.

Geng, G., Zhang, Q., Martin, R. V., Lin, J.-T., Huo, H., Zheng, B., Wang, S., and He, K., 2017. Impact of spatial proxies on the representation of bottom-up emission inventories: A satellite-based analysis. Atmos. Chem. and Phys., 17, 4131-4145.

Giglio, L., J. T. Randerson, and G. R. van der Werf, 2013. Analysis of daily, monthly, and annual burned area using the fourth-generation global fire emissions database (GFED4)", J. Geophys. Res, Biogeosciences. 118, Issue 1, 317-328. Guenther, A. B., Jiang, X., Heald, C. L., Sakulyanontvittaya, T., Duhl, T., Emmons, L. K., and Wang, X., 2012. The Model of Emissions of Gases and Aerosols from Nature version 2.1 (MEGAN2.1): an extended and updated framework for modeling biogenic emissions. Geosci. Model Dev., 5, 1471-1492.

Hudman, R.C., N.E. Moore, R.V. Martin, A.R. Russell, A.K. Mebust, L.C. Valin, and R.C. Cohen, 2012. A mechanistic model of global soil nitric oxide emissions: implementation and space based-constraints. Atmos. Chem. Phys., 12, 7779-7795.

Lin, J., McElroy, M.B., 2010. Impacts of boundary layer mixing on pollutant vertical profiles in the lower troposphere: Implications to satellite remote sensing. Atmos. Environ., 44, 1726-1739.

Rienecker, M. M., Suarez, M. J., Todling, R., Bacmeister, J., Takacs, L., Liu, H.-C., Gu, W., Sienkiewicz, M., Koster, R. D., Gelaro, R., Stajner, I., and Nielsen, J. E.: The GEOS-5 Data Assimilation System – Documentation of Versions 5.0.1, 5.1.0, and 5.2.0, Technical Report Series on Global Modeling and Data Assimilation, NASA Tech. Memo. NASA TM/2008-104606, Vol. 27, 118 pp., 2008.

Zhang L., Liu, L., Zhao, Y.H., Gong, S.L., Zhang, X.Y., D. K. Henze, S. L. Capps, Tzung-May Fu, Zhang, Q., Wang, Y.X., 2015. Source attribution of particulate matter pollution over North China with the adjoint method. Environ Res Lett. 10, 084011.

11. Line 367, ". . .the travelling time (aging time) from LH to HA and from HA to LH were 28 h and 31 h, respectively, which suggested that the BC particle should be coagulated through complex atmospheric processes. Therefore, the new emission inputs along the trajectory enhanced the eBC mass concentration during the transport. . .." How do you infer that "the new emission" enhanced the eBC mass concentration from the previous sentence (longer aging time and coagulation processes) here?

AR: We have revised this part as the following:

*Atmospheric removal of BC occurs in a few days to weeks via wet and dry depositions or contact with surfaces (Bond et al., 2013). In these two cases, there were no precipitation events and the transport time was short (i.e., 28 and 31h), which suggested the less removal rates. Therefore, the new emission inputs along the trajectory enhanced the eBC mass concentration during the transport*

*Previous study found that the BC coagulation with non-refractory materials becomes more significant when the aging timescale was greater than 10 h (Riemer et al., 2004). Chamber studies and field observations also found that the BC absorption enhancement under polluted urban ambient air (Peng et al., 2016, Zhang et al., 2018, Wang et al., 2018c), suggesting the role of aging in modifying BC optical properties. In these two cases, the travelling time (aging time) from LH to HA and from HA to LH was 28 h and 31 h, respectively, which suggested that the BC particle should be coagulated through complex atmospheric processes. Therefore, the $\sigma_{abs}$ was found increased from upwind to downwind site. On the contrary, the AAE values were found decreased during the transport. The AAE is sensitive to the particle size. A lab*

*combustion experiment showed that the particles with smaller diameter from fresh biomass burning have lower AAE value than larger particles (Singh et al., 2016). Simulation also confirmed that the AAE of BC particle decreased with the increasing of its geometric median diameter (Liu et al., 2018b). Therefore, the diameter of BC particle increased during the transportation due to the aging processes supported by the increased absorb coefficients and decreased AAE as discussed above.*

12. Line 370, "...However, slight differences found for BCbb transport: BCbb increased from LH (1.28 ± 0.06 μg m−3) to HA (2.57 ± 0.47 μg m−3), while BCbb decreased from HA (2.37 ± 0.23 μg m−3) to LH (2.14 ± 0.14 μg m−3). . ." What do you mean about this?

AR: In this sentence, we want to express the BC emission difference in these two regions. In case 1, air masses transported from LH (Henan province in north direction) to HA (Hubei province in south direction), both the eBC and BC$_{bb}$ increased However, in Case 2, air masses transported from HA to LH, despite the eBC increased, the BC$_{bb}$ decreased due to less BC emissions in Hubei province than those in Henan province. BC emission inventories also showed this difference (Qin and Xie, 2009, Qiu et al., 2016).

[revised manuscript text omitted]
 as Fig. 3a shows.  Wang et al, (2014b) analyzed ambient BC in an urban site in Xi'an during winter and found the average mass concentration was 8.8 ± 3.7 µg m$^{-3}$, which was higher than that in this study. Compared to other regions,  BC levels in this study were higher than a remote area of Lulang in southeastern part of the Tibetan Plateau (0.31 ± 0.55 µg m$^{-3}$)  (Wang et al.,  (2018) as well as coastal areas such as Hong Kong (1.4 ± 1.1 µg m$^{-3}$) (Wang et al., 2017a) and a rural site in Shenzhen (2.6 ± 1.0 µg m$^{-3}$) (Huang et al., 2012). From BC emission inventory, North and Central China hold higher BC emission intensity (Qin and Xie, 2012; Yang et al., 2017). emission amounts in Hubei and Henan provinces were about 0.6−1.0 g C m$^{-2}$ yr$^{-1}$, which were higher than other regions (Yang et al., 2017). Simulation results also suggested that the near-surface concentrations of BC (6−8 µg m$^{-3}$) in Hubei and Henan were higher than those in south China (4−6 µg m$^{-3}$) during winter (Yang et al., 2017). Compared to the data in other countries (Table S1), BC levels in this study were higher than those in Finland (Hyvärinen et al., 2011), France (Petit et al., 2017b), Ontario (Healy et al., 2017), and south Africa (Chiloane et al., 2017).

For the aerosol absorption properties measured at seven wavelengths by aethalometer, the characteristics (i.e., temporal variation) are generally consistent with each other and the corresponding properties for wavelength at 520 nm is mostly discussed (Zhuang et al., 2015, 2017; Wang et al., 2017b). Then, we only discussed the absorption properties at λ = 520 nm. Figure 3a show the frequency distribution of absorption coefficients (σ$_{abs}$) at three sites. σ$_{abs}$ measured at HA, LH, and WH exhibited a single peak pattern. The average values of σ$_{abs}$ measured at HA, LH and WH were 86.0, 132 and 60.6 Mm$^{-1}$, respectively. Similar to the spatial distribution of BC level, higher σ$_{abs}$ was found in North and Central China, while lower values observed in coastal areas and Tibetan Plateau (Fig. 3b and Table S2).

Figure 3b also shows the average absorption spectra measured at seven wavelengths for different sites. The power law fit was used to calculate the AAE (Zhu et al., 2017). The highest average AAE value was found at LH (1.40), followed by HA (1.32) and WH (1.34). The results indicated that the AAE was different at urban, suburban and rural sites. Generally, the AAE from coal combustion (2.11−3.18) (Sun et al., 2017) and biomass burning (1.85−2.0) (Petit et al., 2017b) were higher than that from traffic sources (0.8−1.1) (Sandradewi et al., 2008; Olson et al., 2015). Therefore, AAE at different sites suggested the different energy consumption structure and more coal or biomass were burned in North China (i.e., LH in this study).

**3.2 Clean days *vs* pollution episodes**

Figure. 54 shows the eBC concentrations under different air quality. It clearly shows that the eBC concentrations increased as the deterioration of air quality. At LH, the average eBC concentrations were 3.39 ± 2.06 µg m$^{-3}$, 8.31 ± 4.55 µg m$^{-3}$ and 13.0 ± 4.59 µg m$^{-3}$, respectively when the air quality was clean, light polluted and heavy polluted. The average values of eBC increased by 163%, 139%, 96.2%, 51.8% and 26.4% at SX, XY, LH, HA and WH, respectively from clean to pollution. The eBC enhancement  along with the  air quality deterioration was also reported elsewhere (Wang et al., 2014b, c; Liu et al., 2016, 2018a). At LH and HA, the enhancement of eBC level from clean to pollution period was due to both the elevated BC emissions from biomass burning (BC$_{bb}$) and fossil fuel combustion (BC$_{ff}$) (Fig. 54b and 54c). The BC$_{ff}$ accounted for a higher contribution to eBC and the percentages of BC$_{bb}$ to eBC decreased during the haze episodes (Fig. 4d). At WH, both the concentration and percentage of BC$_{bb}$  decreased from clean to pollution, which suggested that more BC$_{ff}$ was emitted during haze episodes. This finding was different with previous study conducted in Beijing that the absolute concentration and percentage of  biomass burning and coal combustion   increased from clean to pollution episodes (Liu et al., 2018a). The differences suggested that the control of fossil fuel combustion (vehicle emissions) instead of coal or biomass burning should be taken priority during the haze episodes at WH. While it should give priority to biomass and coal combustion control in North China to prevent air pollution.

Additionally, the aerosol optical properties (σ$_{abs}$ and AAE) also exhibited different levels under different air quality. Similar to eBC levels, the σ$_{abs}$  elevated by 11.7−254% as the air quality switched from clean to pollution (Fig. 54e). Our observation (Fig. S6) and previous study found that there are more secondary aerosols (i.e., sulfate, nitrate) during the pollution episodes (Huang et al., 2014). The increased secondary aerosols would be more adsorbed on  BC particle and therefore, the BC absorption  enhanced via the lens effects of these coated materials (Jacobson, 2000; Moffet and Prather, 2009). On the contrary, the AAE showed higher values during clean days when compared to pollution episodes (Fig. 54f). The AAE decreasing  from clean to polluted days was also reported elsewhere (Zhang et al., 2015b) and it can be partly attributed to the source variation. The AAE for biomass burning is about 2.0 while the AAE for fossil fuel combustion is about 1.0 (Sandradewi et al., 2008). Higher AAE values during clean days suggested that more BC  may be from biomass burning and lower AAE indicated the dominance of fossil fuel combustion during the pollution period (Fig. 54c). The AAE is also sensitive to other factors such as the particle size. Previous studies suggested that the particle diameter and number concentration increased from clean to pollution episodes due to several factors such as coagulation, hygroscopic growth, emissions, meteorological conditions, i.e., planetary boundary layer and wind speed (Guo et al., 2014; Zhang et al., 2017). These studies suggested that the particle diameter is generally larger during pollution days. Furthermore, the lab combustion and numeric simulation proved that BC particle with larger geometric median diameter had lower AAE value (Singh et al., 2016; Liu et al., 2018b). Therefore, lower AAE was observed during pollution episodes in this study.

Figure 6 and Fig. S7 shows the diurnal variations of eBC and absorption coefficients under different air quality. The diurnal cycles of black carbon and absorption showed similar variation patterns. he BC mass concentrations were discussed here. 
[revised manuscript text omitted]
. For instance, at WH, high levels of eBC (> 2.5 μg m$^{-3}$) were found from south direction, while the source regions with high level eBC (> 3 μg m$^{-3}$) switched to northeast direction when the air quality was worsened. Figure 12 shows the semiquantitative results of transportation contribution results during clean and pollution episodes. At the boundary sites (HA, SX and XY), BC was mainly from south direction (accounting for 46.0−58.2%) when the air quality was clean, and it was mainly from northeast/northwest directions (51.2−76.5%) when the air quality getting worse. At SE-NCP site (LH), BC was dominantly from south direction (47.8%) during pollution episodes. At CC site (WH), BC was mainly from northeast direction (49.3−71.1%). These results suggested that northwest and northeast directions were the main transport pathways of air pollutants reaching to WH during the pollution episodes. Furthermore, to control local emissions during haze episodes, the emission sources, i.e., industry plant and open biomass burning in the upwind direction should also be controlled to prevent the further deterioration of air quality in downwind areas.

~~The CWT plots of $BC_{bb}$ and $BC_{ff}$ showed similar distribution with eBC. At LH, CWT results of eBC, $BC_{bb}$ and $BC_{ff}$ showed that high levels of eBC were mainly from south direction during clean days. When the air quality degraded to pollution, the air masses from south and east directions both contributed to the high eBC (i.e., > 5 μg m$^{-3}$ for BC) at LH. CWT results at WH showed that the southeast direction was the dominant source regions of eBC, $BC_{bb}$ and $BC_{ff}$ during clean days, while the source regions switched to northeast direction when the air quality changed into pollution.~~

~~To give quantified results of which cluster had greater contribution to eBC level at the receptor sites, the percentage contributions of each cluster reaching at the five sites under different air pollution situation are summarized in Figure 11. Trajectories from south was the main transport pathways reaching at HA, which accounted for 49.6% for clean days and therefore, the highest average eBC level (6.33±1.61 µg m⁻³) was found from south direction during the clean days. However, the percentage contribution for the south cluster contributed least (21.4%) to the total air masses, but it had the highest eBC level (7.89±2.59 µg m⁻³) among the three clusters during the pollution episode. At LH, despite the lowest ratio of trajectories were found from south (16.7%) and northwest (21.6%), respectively for clean and pollution days, these two clusters had the highest eBC levels. At WH, the cluster with the highest percentages also had the highest BC levels. For instance, northeast direction was the primary pathway of BC reaching at WH and the highest average eBC value was also found from this direction. In summary, at the boundary sites (HA, SX and XY), BC was mainly from south direction (accounting for 46.0–58.2%) when the air quality was clean and it was mainly from northeast/northwest directions (51.2–76.5%) when the air quality getting worse. At SE NCP site (LH), BC was dominantly from south direction (47.8%) during pollution episodes in this study. At CC site (WH), BC was mainly from northeast direction (49.3–71.1%). These results suggested that northwest and northeast directions were the main transport pathways of air pollutant reaching to WH during the pollution episodes. Furthermore, in addition to control local emissions during haze episodes, the emission sources, i.e., industry plant and open biomass burning in the upwind direction should also be controlled to prevent the further deterioration of air quality.~~

**3.6 Case studies for BC properties variation during transportation**

To explore the BC variations (i.e., mass concentration, sources and AAE) during the transportation, we chose two cases. LH and HA were selected as the study sites due to the same instrument deployment (AE33) and they are representative of SE-NCP and CC. BC transportation from HA to LH and from LH to HA were both considered. Figure 13a show the hourly backward trajectories reaching at HA on 2018-1-12 and the trajectory at 13:00  (GMT) (black line) was found passing through LH  and the travelling time was about 28 h. Therefore, the eBC mass concentration (including $BC_{ff}$ and $BC_{bb}$),  and AAE at the upwind site LH on 8:00 2018-1-11 (GMT)  downwind site HA on 13:00, 2018-1-13  (GMT) were compared (Fig. 13b). In case 1, during the air transport from LH to HA, eBC, $BC_{ff}$  $BC_{bb}$ significantly increased ($p < 0.01$). The  BC absorption enhancement from $25.6 \pm 0.81$ Mm⁻¹ (LH) to $61.8 \pm 12.5$ Mm⁻¹ (HA) was also observed, while the AAE significantly decreased from $1.49 \pm 0.02$ to $1.42 \pm 0.02$ ($p < 0.01$). Similarly, in case 2, the air masses reaching at LH on 7:00, 2018-1-13  (GMT) were also passing through HA (black line) about 31 h ago  (Figure 13c). The eBC $BC_{ff}$ and $\sigma_{abs}$ increased from upwind (HA) to downwind (LH), while $BC_{bb}$ and AAE decreased from $2.37 \pm 0.23$ µg m⁻³ and $1.43 \pm 0.02$ to $2.14 \pm 0.14$ µg m⁻³ and $1.32 \pm 0.01$, respectively (Fig. 13d). The eBC mass concentrations  enhanced during the transportation regardless of the transport direction was from CC to NCP or from NCP to CC. Atmospheric removal of BC occurs in a few days to weeks via wet and dry depositions or contact with surfaces (Bond et al., 2013). In these two cases, there were no precipitation events and the transport time was short (i.e., 28 and 31h), which suggested the less removal rates. Therefore, the new emission inputs along the trajectory enhanced the eBC mass concentration during the transport. However, slight differences were found for $BC_{bb}$ transport: $BC_{bb}$ increased from north direction (LH: $1.28 \pm 0.06$ µg m$^{-3}$) to south direction (HA: $2.57 \pm 0.47$ µg m$^{-3}$), while $BC_{bb}$ decreased from HA ($2.37 \pm 0.23$ µg m$^{-3}$) to LH ($2.14 \pm 0.14$ µg m$^{-3}$). The difference suggested that there were more intensive biomass burning emissions in Henan than Hubei province, which was also verified by the BC emission inventory (Qin and Xie, 2012; Qiu et al., 2016).

Previous stud found that the BC coagulation with non-refractory materials becomes more significant when the aging timescale was greater than 10 h (Riemer et al., 2004). Chamber stud and field observations also found that the BC absorption enhancement under polluted urban ambient air (Peng et al., 2016, Zhang et al., 2018, Wang et al., 2018c), suggesting the role of aging in modifying  BC optical properties. In these two cases, the travelling time (aging time) from LH to HA and from HA to LH were 28 h and 31 h, respectively, which suggested that the BC particle should be coagulated through complex atmospheric processes. Therefore, the $\sigma_{abs}$ was found increased from upwind to downwind site. On the contrary, the AAE values were found decreased during the transport. The AAE is sensitive to the particle size. Ahe lab combustion experiment showed that the particles with smaller diameter from fresh biomass burning have lower AAE value than larger particles (Singh et al., 2016). Simulation also confirmed that the AAE of BC particle decreased with the increasing of its geometric median diameter (Liu et al., 2018b). Therefore, the diameter of BC particle increased during the transportation due to the aging processes supported by the increased absorb coefficients and decreased AAE as discussed above~~The same result was also found through numerical simulation (Liu et al., 2018b). Additionally, chamber study of diesel soot particles coated with secondary organic compound also found that the AAE decreased from 1.13 to 0.8 (Schnaiter, 2005). The decreasing of AAE of BC particles during the transport indicated that BC was coated by other materials.the new emission inputs along the trajectory enhanced the eBC mass concentration during the transport.However, slight differences found for $BC_{bb}$ transport: $BC_{bb}$ increased from LH ($1.28 \pm 0.06$ µg m$^{-3}$) to HA ($2.57 \pm 0.47$ µg m$^{-3}$), while $BC_{bb}$ decreased from HA ($2.37 \pm 0.23$ µg m$^{-3}$) to LH ($2.14 \pm 0.14$ µg m$^{-3}$).~~

**4 Summary**

[revised manuscript text omitted]

**Figure**  4 Frequency distribution of absorption coefficients ($\sigma b_{abs}$) at 520 nm wavelength (left panel) and power fit of $\sigma b_{abs}$ at seven wavelengths (right panel) for HA, LH, and WH.

[Figure]

**Figure 4 5** Box (25−75th percentiles) and whisker (5−95th percentiles) plots of eBC concentrations (a), $BC_{bb}$ (b), $BC_{ff}$ (c), percentages of $BC_{bb}$ (d), aerosol absorption coefficients (e), and absorption Ångström exponent (AAE) under different air pollution situation. The blue, orange and black color represent the clean (PM$_{2.5}$ <75 μg m$^{-3}$), light pollution (75< PM$_{2.5}$ <250 μg m$^{-3}$) and heavy pollution conditions (PM$_{2.5}$ >250 μg m$^{-3}$), respectively. The data number for the different air quality could be found in the supplementary file (Table S4).

[Figure]

**Figure**  6 Diurnal variations of eBC under different air pollution situations (blue: clean; orange: light polluted; dark: heavy polluted) at the five observation sites. The solid lines are the average values and the filled ribbons are 95th confidential intervals of the average value.

[Figure]

**Figure** 6 7 Ratios of BC/PM$_{2.5}$ (a) and BC/CO (b) in this study and previous researches.

[a] Chow et al., (2011); [b] Zhang et al., (2009); [c] Dhammapala et al., (2007); [d] Cao et al., (2008); [e] Andreae and Merlet, (2001); [f] Streets et al., (2003); [g] Westerdahl et al., (2009); [h] Liu et al., (2018a); [i] Park et al., (2005) ; [j] Vermal et al., (2010); [k] Kondo et al., (2006); [l] Pan et al., (2011).

[Figure]

**Figure**  8  Conditional bivariate probability function (CBPF) plots of $BC_{bb}$ (left panel) and $BC_{ff}$ (right panel) at HA, LH and WH.

[Figure]

**Figure** 8 9 Time series of surface transport intensity for BC at the five observation sites. Positive values for HA and LH indicated the transport direction was from north to south and negative values indicated the transport direction was from south to north. Positive values for SX, WH and XY indicated the transport directions were from northwest to southeast and negative values indicated the transport directions were from southeast to northwest.

[Figure]

**Figure 9 10** Concentration-weighted trajectory (CWT) plots of BC$_{bb}$ (left panel) and BC$_{ff}$ (right panel) at HA, LH and WH during the whole observation site. The white dote represents the observation site.

[Figure]

**Figure** 10 11 Concentration-weighted trajectory (CWT) plots of eBC, BC*bb* and BC*ff* during clean and pollution episodes at HA, LH and WH. The white dot represents the observation site.

[Figure]

**Figure 121** Cluster results of air masses reaching at five sites (inner pie plots) and the eBC percentage contributions from different clusters (extern pie plot) during the clean days (up panel) and pollution episodes (bottom panel). NW, NE and S mean the northwest, northeast and south direction clusters as shown in Figure 1.

[Figure]

**Figure 132** Case studies of BC variation during the transportation from upwind to downwind direction. a (case 1): Hourly backward trajectories (grey line) reaching at HA on 2018-1-12 and the trajectory at 13:00 (GMTUTC) (black line) was found passing through LH about 28 hours ago. c (case 2): Trajectory reaching at LH on 2018-1-13 07:00 (GMTUCT) (black line) was found passing through HA about 31 hours ago. Box (25-75th percentiles) and whisker (5-95th percentiles) plots of eBC, BC$_{ff}$, BC$_{bb}$ $\sigma_{abs}$, and AAE variations during the transport from LH to HA (b) and from HA to LH (d).